# Single-cell RNA sequencing defines developmental progression and reproductive transitions of *Pneumocystis carinii*

Aaron W. Albee,[1,2] Steven G. Sayson,[1,2] Alan Ashbaugh,[1,2] Nicholas J. Wolf,[1,2] Aleksey Porollo,[3,4] George Smulian,[1,2] Melanie T. Cushion[1,2]

**ABSTRACT** *Pneumocystis* species are host-obligate fungal pathogens that cause severe pneumonia in immunocompromised individuals. Despite their clinical importance, their life cycle remains poorly understood, in part because *Pneumocystis* depends on the host environment for most nutrients and requires sexual reproduction for survival, which occurs exclusively *in vivo*. This study presents the first single-cell RNA sequencing (scRNA-seq) atlas of *Pneumocystis carinii*, generated from isolated organisms recovered from the bronchoalveolar lavage fluid of infected rats to map the life cycle of *P. carinii*. Transcriptomes from 87,716 cells were analyzed using the 10× Genomics platform, revealing 13 transcriptionally distinct clusters representing key developmental stages, including biosynthetically active trophic forms, mating-competent intermediates, and asci undergoing sporulation. These states were characterized by expression of MAPK signaling components, β-glucan-modifying enzymes, and spore-associated genes, respectively. The scRNA-seq data support previous evidence that these host-obligate fungi undergo sexual reproduction and provide new insights into the gene expression patterns associated with different life cycle phases. Biomarkers associated with ascus formation identified by scRNA-seq were validated by RT-qPCR, showing decreased expression levels in ascus-depleted populations treated with anidulafungin, a drug that halts ascus formation. More broadly, this approach provides a strategy for studying the full life cycles of fungal pathogens that cannot be continuously cultured.

**IMPORTANCE** *Pneumocystis* species (spp.) are clinically significant fungal pathogens that cannot be sustainably cultured *in vitro* due to their host-obligate nature. This longstanding limitation has impeded progress in understanding their life cycle and identifying therapeutic vulnerabilities. Here, we apply scRNA-seq to *P. carinii* isolated directly from infected rat lungs, generating the first transcriptional map of its developmental progression. Our results define discrete gene expression states associated with trophic growth, mating activation, and ascus formation and provide transcriptional evidence for a structured life cycle, clarifying key developmental transitions and identifying potential regulatory targets for therapeutic intervention. Importantly, this study demonstrates that scRNA-seq can resolve the developmental biology of host-restricted fungal pathogens that cannot be cultured *in vitro*. This approach offers a generalizable framework for investigating other unculturable or obligate microbial pathogens directly within their native host environments, where traditional experimental tools are limited.

**KEYWORDS** *Pneumocystis*, *Pneumocystis carinii*, fungal pathogen, infectious disease, 10× Genomics, single-cell RNA sequencing, *Pneumocystis* pneumonia, opportunistic fungi

P̲neumocystis species are host-obligate fungal pathogens that cause life-threatening pneumonia in immunocompromised individuals. The human-specific species,

**Peer Reviewer** Alexandre Alanio, Institut Pasteur, Paris, France

Address correspondence to Melanie T. Cushion, cushiomt@ucmail.uc.edu.

The authors declare no conflict of interest.

See the funding table on p. 22.

*Pneumocystis jirovecii*, is responsible for *P. jirovecii* pneumonia (PjP) in patients with HIV/AIDS, organ transplants, or hematologic malignancies (1–3). These fungi adhere to type I pneumocytes in the alveolar lining, where their proliferation and the associated inflammatory response can lead to acute respiratory distress syndrome (ARDS; 4–6). Despite their clinical importance, the basic biology of *Pneumocystis* spp., including reproduction, transmission, and persistence in the lungs, remains poorly understood.

*Pneumocystis* spp. lack an environmental stage and cannot survive without the host lung (7), and this constraint has made long-term culture and genetic manipulation unfeasible (8). Their mostly strict host specificity further complicates research; for example, *P. jirovecii* infects only humans, precluding direct study. To overcome this limitation, related species such as *P. carinii* (in rats) and *P. murina* (in mice) serve as experimental models for investigating infection dynamics, host-pathogen interactions, and life cycle progression (9). Genomic analyses show that *Pneumocystis* spp. have lost numerous biosynthetic and metabolic pathways common to most fungi (10, 11), including those for amino acid, carbohydrate metabolism, and lipid synthesis. These fungi complete their entire life cycle within the mammalian lung, relying on the host for nutrients and undergoing sexual reproduction *in situ*. As such, their biology dictates both their pathogenic strategy and the tools available for their study.

The life cycle has been largely determined by photomicrographs, which indicated an asexual phase via binary fission and a sexual phase resulting in the production of asci, structures essential for transmission to a new host (12, 13). Prior work from our group has demonstrated that *P. murina* relies on sexual reproduction for survival and transmission (14, 15). Ascus formation, a key step in the sexual cycle, is required for completion of the life cycle and is inhibited by β-1,3-glucan synthesis inhibitors, such as anidulafungin. A better understanding of stage-specific gene expression is essential for identifying developmental regulators and potential vulnerabilities in the organism's life cycle.

This study defines the first single-cell transcriptomic map of *P. carinii*, resolving gene expression patterns across its life cycle. It addresses three major challenges in *Pneumocystis* spp. research: the inability to culture the organism *ex vivo*, the essential role of sexual reproduction in its biology, and the lack of molecular detail regarding the sequence and coordination of life cycle stages. By adapting and optimizing a protocol for fungal cell isolation from bronchoalveolar lavage fluid (BALF), we preserved the transcriptomic profiles of individual cells, offering a first glimpse of *P. carinii* development within the host. This stage-resolved transcriptional framework lays the groundwork for targeted investigations of developmental regulation, therapeutic disruption of transmission, and comparative studies of other host-adapted fungal pathogens.

## RESULTS

### Optimization of *Pneumocystis carinii* single-cell preparation for scRNA-seq

To facilitate single-cell transcriptomic profiling, a preparation protocol for *P. carinii* was developed to be compatible with the 10× Genomics platform (Fig. S1A). Given that *P. carinii* populations are composed primarily of trophic forms (~90%) and only ~10% asci (16), density gradient centrifugation was employed to enrich for asci (Fig. S1B) (17) and to reduce host cell contamination. In unseparated BALF, asci represented a minority population and were underrepresented in bulk analyses. Enrichment was necessary to capture enough asci and their precursor stages to resolve the transcriptional trajectories and gene expression patterns associated with sexual reproduction and ascus formation. Following Ficoll separation, trophic forms were primarily recovered in the 2%–4% gradient layers, while asci were enriched in the 6%–12% layers, comprising 75%–95% of the fungal cells within these fractions (Fig. S1C). The ascus-to-trophic form ratio increased progressively from approximately 3:1 in the 6% layer to over 17:1 in the 12% layer (Fig. S1D). Recovery of asci from the most enriched fractions reached up to 60%, and gradient performance was consistent across biological replicates. Organisms were obtained by bronchoalveolar lavage rather than lung homogenization, which further

reduced the number of host cells. Post-enrichment viability consistently exceeded 80% (Fig. S2A). Lysis conditions were adjusted by adding zymolyase to improve disruption of the 1,3-β-D-glucan-rich ascus wall (Fig. S2B) (18). Two preliminary experiments informed adjustments to ensure sufficient sequencing depth for downstream analysis (Tables S1 to S3).

## Single-cell RNA-seq reveals 13 distinct transcriptional states across the *P. carinii* life cycle

scRNA-seq of *P. carinii* isolated from bronchoalveolar lavage fluid (BALF) of infected rat lungs recovered 87,502 cells (Table S3) and identified 13 transcriptionally distinct clusters (C1-C13), each representing a specific stage in the fungal life cycle (Fig. 1A). Pseudotime trajectory analysis placed clusters C1–C8 at earlier stages and clusters C9–C13 at later stages (Fig. 1B and C), with pseudotime values transitioning from low in C1–C8 to high in C9–C13. The heatmap of gene expression (Fig. 1D) shows the top five highly expressed genes in each cluster, with regulation indicated by a color gradient: upregulated genes in pink, downregulated genes in blue, and non-differentially expressed genes in white. GO term enrichment analysis (Fig. 2A) further supported these findings, revealing distinct biological processes associated with each cluster and showing three potential life cycle phases, described in detail below. Together, these data show a structured progression through the *P. carinii* life cycle, with transitions between trophic, mating, and sexual reproduction states.

Clusters C1–C4 correspond to early trophic stages characterized by elevated metabolic and translational activity, with little indication of mitotic division. In Cluster C1, upregulation of *ptf1* (phosphoric monoester hydrolase) and *ubc11* (ubiquitin conjugation) (Table 1) reflects active metabolic processes, including phosphate metabolism, signal transduction, metabolic regulation, and protein turnover. The genes *pmp3* (plasma membrane proteolipid involved in ion homeostasis) and *mug113* (meiotic regulator) likely represent remnants of expression from Cluster C13, linking early and late stages of the life cycle. Downregulation of the mitotic gene *stg1* (cell cycle regulation, G2/M) and ascus-related cell wall genes (*gas1*, *och1*) (Table 2) supports the suppression of mitosis and ascus-related genes upregulated in Clusters C11–C13. GO term enrichment (Table 3) highlights processes such as cytoplasmic translation, biosynthesis, and metabolic processes, suggesting a metabolically active state, while sexual reproduction-related pathways remain minimally expressed, likely reflecting residual mRNA from Cluster C13.

In Cluster C2, several transcripts encoding ribosomal proteins (Table 1) are upregulated, along with genes involved in protein synthesis, transportation, and cellular metabolism (*syb1*, *sam1*, and *erg6*), reflecting a cellular state primed for biosynthetic activity and biomass accumulation. Concurrently, the downregulation of genes (Table 2) essential for mitotic progression and chromosome segregation (*spc24*, *spc25*, *nuf2*, *slp1*, and *spo12*), as well as those involved in cell remodeling (*eng1* and *adg3*), suggests a transient suppression of cell division processes. Together, these expression patterns are consistent with translational readiness and metabolic investment over progression through mitosis or meiosis. GO term enrichment (Table 3) for translation and metabolic processes underlines the continued biosynthetic activity in these cells.

Cluster C3 shows upregulation of genes involved in lipid metabolism (*ole1*), mitochondrial transport, and heme biosynthesis (*odc1* and *hem1*), ribosome biogenesis (*kri1*), and transcriptional and chromatin regulation (*cti6*, *loz1*, and *cdt2*), representing a shift toward metabolic remodeling and preparatory gene expression (Table 1). Upregulation of *krp1* (pheromone precursor maturation) suggests a readiness for mating. Although *cut14*, involved in mitosis, is upregulated, it also plays a role in chromatin organization, a function that can occur outside mitosis in response to stimuli. Simultaneously, the suppression of genes critical for mitotic exit and cytokinesis (*slp1*, *ace2*, *eng1*, *spo12*, and *stg1*) (Table 2), as well as chromatin structure (*hht1*), suggests a transient delay or checkpoint in the cell cycle. The downregulation of mitotic genes is consistent with the absence of mitotic division, supporting the roles of *cut14* and *cti6* in

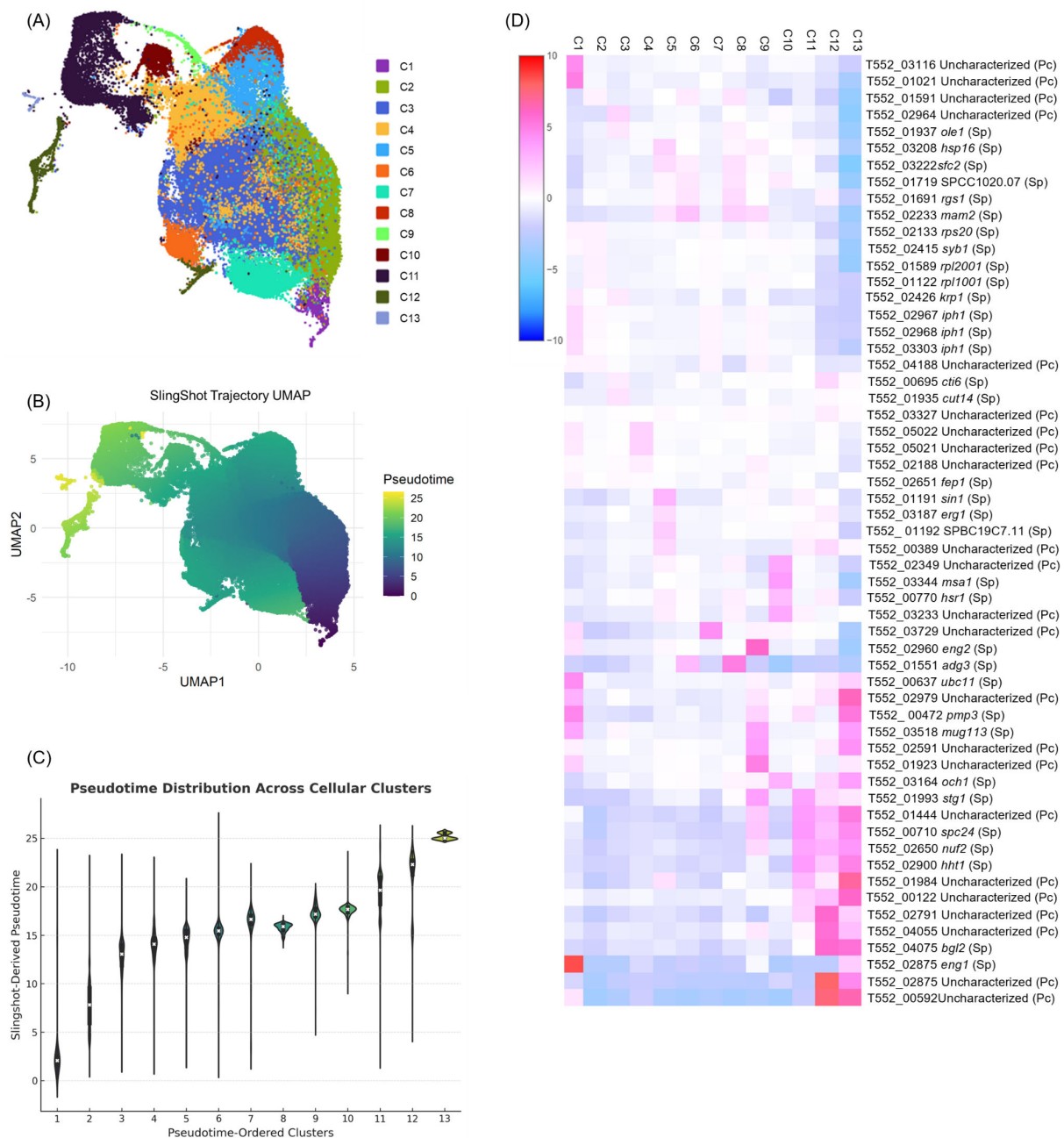

**FIG 1** scRNA sequencing identifies transcriptionally distinct clusters in *P. carinii*. (A) Uniform Manifold Approximation and Projection (UMAP) of *P. carinii* cells, clustered into 13 transcriptionally distinct populations (C1–C13). (B) Pseudotime progression derived from Slingshot analysis, illustrating the inferred developmental trajectory from early clusters (dark purple) to late clusters (yellow). (C) Violin plot showing the distribution of Slingshot-derived pseudotime values (Y-axis) for cells grouped across transcriptional clusters (x-axis). (D) Heatmap showing the log-fold regulation of the top five differentially expressed genes (rows) across the 13 transcriptional clusters (columns). Each cell represents the average log-fold change in gene expression within each cluster.

chromatin structure outside mitosis. GO term enrichment (Table 3) highlights metabolic and biosynthetic processes, reinforcing the cluster's focus on metabolism.

Cluster C4 reflects a cellular adaptation to environmental or physiological stress, marked by upregulation of stress-response and membrane maintenance pathways and concurrent downregulation of mitotic progression and chromatin-related features (Table 2). Specifically, the increased expression of *sin1* and *hsp16*, along with chaperones like *ssa2*, suggests activation of pathways that protect protein integrity and

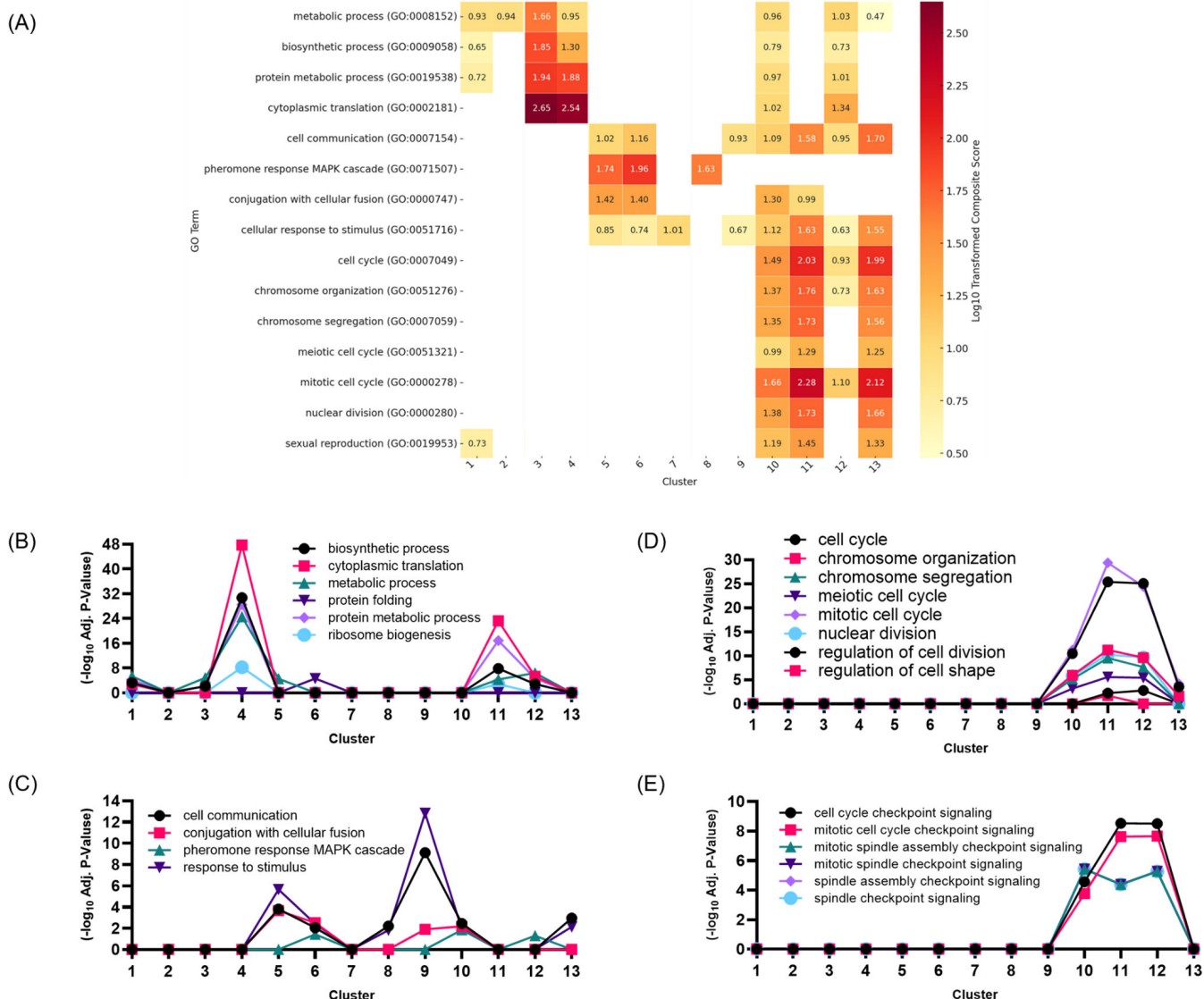

**FIG 2** Transcriptional signatures from GO term enrichment define three distinct developmental transitions in *P. carinii*'s life cycle. (A) GO biological process enrichment heatmap showing 14 GO terms significantly enriched across transcriptional clusters. Enrichment is represented as a –log$_{10}$-transformed adj. *P*-value. Clusters are along the x-axis, and GO terms are listed on the y-axis. Dark red indicates higher enrichment (log$_{10}$ fold change ≥2.5), while pale yellow indicates lower enrichment (~0.5). (B) Line graph illustrating the enrichment of metabolic, biosynthetic, and protein metabolic processes across transcriptional clusters ordered by pseudotime. (C) Line graph showing the enrichment of sexual reproduction and signaling processes, including pheromone response and the MAPK cascade, across the transcriptional clusters. (D) Line graph highlighting enrichment of cell cycle-related processes, such as chromosome organization, segregation, and nuclear division, in late-stage clusters. (E) Line graph showing the enrichment of checkpoint-related pathways (e.g., mitotic spindle checkpoint signaling), which regulate cell division, particularly in late-stage development.

membrane function. Simultaneously, the induction of lipid biosynthesis genes (*erg1*, *pss1*) and ribosome biogenesis factors (*ipi1*) indicates an investment in cellular repair and metabolic recalibration. By contrast, the downregulation of genes essential for chromatin structure (*hht1*), cell cycle progression (*spo12*, *spc24*, and *spc25*), and cell wall remodeling (*eng1*) implies a deliberate slowing or arrest of cell division. This coordinated shift likely enables the cell to prioritize homeostasis and survival over proliferation, a strategic response to ensure long-term viability under suboptimal conditions. GO term enrichment (Table 3) for translation and biosynthesis reinforces the metabolically active state of these cells. In Cluster C3 and C4, the downregulation of *eng1* persisted while

**TABLE 1** Top differentially upregulated genes per cluster

| Cluster | Feature name | Log2 FC[a] | Adj. P-value | Homolog[b] | Function |
|---|---|---|---|---|---|
| 1 | T552_01021 | 5.054589633 | 0.00E+00 | –[c] | Uncharacterized protein |
| 1 | T552_00472 | 4.763150831 | 2.36E-234 | pmp3 (Sp) | Plasma membrane proteolipid |
| 1 | T552_03116 | 4.574652528 | 0.00E+00 | – | Uncharacterized protein |
| 1 | T552_00637 | 4.52711577 | 0.00E+00 | ubc11 (Sp) | Protein polyubiquitination |
| 1 | T552_04113 | 4.23013256 | 5.50E-266 | ptf1 (Sp) | Phosphoric monoester hydrolase |
| 1 | T552_02714 | 4.168804739 | 1.61E-171 | SPAC15E1.02c | Implicated in phosphate metabolism |
| 1 | T552_00307 | 3.951337422 | 1.47E-249 | pit1 (Sp) | Intracellular signal transduction |
| 1 | T552_04058 | 3.945169362 | 6.93E-125 | – | Uncharacterized protein |
| 1 | T552_03518 | 3.531502672 | 7.85E-110 | mug113 (Sp) | Meiotic cell cycle, stress response |
| 1 | T552_02877 | 3.478718626 | 6.31E-106 | – | Uncharacterized protein |
| 2 | T552_01589 | 0.574669886 | 1.01E-85 | rpl2001 (Sp) | Cytoplasmic translation |
| 2 | T552_02133 | 0.554834612 | 1.57E-81 | rps20 (Sp) | Cytoplasmic translation |
| 2 | T552_01122 | 0.549402496 | 4.98E-79 | rpl1001 (Sp) | Cytoplasmic translation |
| 2 | T552_02415 | 0.53650342 | 2.46E-75 | syb1 (Sp) | Membrane formation |
| 2 | T552_02084 | 0.511177802 | 1.09E-67 | rpl29 (Sp) | Cytoplasmic translation |
| 2 | T552_02499 | 0.509829859 | 1.29E-54 | sam1 (Sp) | One-carbon metabolic process |
| 2 | T552_03167 | 0.507436461 | 9.28E-69 | erg6 (Sp) | Ergosterol biosynthetic process |
| 2 | T552_01405 | 0.49651809 | 1.63E-65 | rps2 (Sp) | Cytoplasmic translation |
| 2 | T552_03524 | 0.489793114 | 1.51E-64 | rpl2702 (Sp) | Cytoplasmic translation |
| 2 | T552_00095 | 0.486583259 | 2.83E-63 | rpl1602 (Sp) | Cytoplasmic translation |
| 3 | T552_02964 | 1.672841554 | 0.00E+00 | – | Uncharacterized protein |
| 3 | T552_01935 | 1.118936806 | 8.00E-265 | cut14 (Sp) | Cell division |
| 3 | T552_01937 | 0.998807601 | 3.79E-209 | ole1 (Sp) | Unsaturated fatty acid biosynthetic process |
| 3 | T552_02426 | 0.875817909 | 6.15E-139 | krp1 (Sp) | Peptide mating pheromone maturation |
| 3 | T552_00695 | 0.796786211 | 1.87E-124 | cti6 (Sp) | Chromatin organization |
| 3 | T552_03211 | 0.695465246 | 7.22E-106 | odc1 (Sp) | Malate-aspartate shuttle |
| 3 | T552_03516 | 0.678649554 | 7.10E-82 | hem1 (Sp) | Heme biosynthetic process |
| 3 | T552_04054 | 0.651253062 | 3.57E-81 | kri1 (Sp) | Endonucleolytic cleavage, ribosome biogenesis |
| 3 | T552_00385 | 0.554681933 | 2.82E-51 | loz1 (Sp) | Intracellular zinc ion homeostasis |
| 3 | T552_00870 | 0.543392738 | 4.76E-52 | cdt2 (Sp) | Protein ubiquitination |
| 4 | T552_01191 | 2.917007262 | 0.00E+00 | sin1 (Sp) | TORC2 signaling |
| 4 | T552_01192 | 2.008680328 | 0.00E+00 | SPBC19C7.11 | Intracellular copper/iron ion homeostasis |
| 4 | T552_03208 | 1.809148786 | 0.00E+00 | hsp16 (Sp) | Protein folding |
| 4 | T552_00389 | 1.582097811 | 3.08E-216 | – | Uncharacterized protein |
| 4 | T552_03187 | 1.446401181 | 8.65E-211 | erg1 (Sp) | Ergosterol biosynthetic process |
| 4 | T552_03222 | 1.361717987 | 1.73E-183 | sfc2 (Sp/Sc) | Membrane transport role |
| 4 | T552_00013 | 1.315273833 | 6.02E-199 | pss1 (Sp) | Protein refolding, phosphatidylserine synthase |
| 4 | T552_01984 | 1.209908071 | 4.16E-53 | – | Uncharacterized protein |
| 4 | T552_01031 | 1.186775916 | 9.12E-158 | ipi1 (Sp) | rRNA processing |
| 4 | T552_02628 | 1.062735529 | 5.43E-135 | ssa2 (Sp) | Protein refolding |
| 5 | T552_01551 | 2.756218897 | 0.00E+00 | adg3 (Sp) | Cell wall organization |
| 5 | T552_02233 | 2.39902915 | 0.00E+00 | mam2 (Sp) | Pheromone-dependent signal transduction |
| 5 | T552_01691 | 1.097251989 | 4.58E-84 | rgs1 (Sp) | Pheromone response MAPK cascade |
| 5 | T552_01719 | 0.980152626 | 1.51E-68 | SPCC1020.07 | Pyrimidine nucleoside salvage |
| 5 | T552_02827 | 0.805062949 | 3.62E-49 | – | Uncharacterized protein |
| 5 | T552_02142 | 0.766119205 | 1.13E-37 | spk1 (Sp) | Pheromone response MAPK cascade |
| 5 | T552_03222 | 0.729960553 | 5.99E-37 | sfc2 (Sp/Sc) | Membrane transport role |
| 5 | T552_02205 | 0.712498774 | 1.17E-36 | tef3 (Sp) | Cytoplasmic translation |
| 5 | T552_00432 | 0.530213164 | 1.61E-17 | – | Uncharacterized protein |
| 5 | T552_01049 | 0.523322865 | 1.32E-18 | fil1 (Sp/Sc) | Amino acid starvation response |
| 6 | T552_04188 | 0.465506839 | 1.56E-10 | – | Uncharacterized protein |
| 6 | T552_03303 | 0.455838393 | 6.38E-11 | iph1 (Sp) | Peptide pheromone maturation, |

(*Continued on next page*)

TABLE 1 Top differentially upregulated genes per cluster (*Continued*)

| Cluster | Feature name | Log2 FC[a] | Adj. *P*-value | Homolog[b] | Function |
|---|---|---|---|---|---|
| 6 | T552_02968 | 0.440159212 | 4.34E-09 | *iph1* (Sp) | Peptide pheromone maturation, |
| 6 | T552_02967 | 0.438567685 | 4.93E-09 | *iph1* (Sp) | Peptide pheromone maturation, |
| 6 | T552_03304 | 0.433144128 | 4.24E-09 | *iph1* (Sp) | Peptide pheromone maturation, |
| 6 | T552_00353 | 0.418195568 | 2.03E-09 | – | Uncharacterized protein |
| 6 | T552_01110 | 0.40944558 | 1.13E-08 | *krp1* (Sp) | Peptide mating pheromone maturation |
| 6 | T552_03424 | 0.383719932 | 9.46E-08 | *krp1* (Sp) | Peptide mating pheromone maturation |
| 6 | T552_02821 | 0.377083206 | 4.86E-07 | *cpy1* (Sp) | Zymogen activation, Carboxypeptidase |
| 6 | T552_03346 | 0.370215564 | 6.52E-07 | *bip1* (Sp) | ERAD pathway |
| 7 | T552_05021 | 1.804017359 | 9.17E-133 | – | Uncharacterized protein |
| 7 | T552_05022 | 1.799539012 | 3.05E-131 | – | Uncharacterized protein |
| 7 | T552_02188 | 0.659271633 | 2.00E-10 | – | Uncharacterized protein |
| 7 | T552_02651 | 0.194448839 | 4.40E-02 | *fep1* (Sp) | Intracellular iron ion homeostasis |
| 8 | T552_01551 | 5.067200642 | 0.00E+00 | *adg3* (Sp) | Probable secreted beta-glucosidase |
| 8 | T552_02233 | 1.879023953 | 3.70E-51 | *mam2* (Sp) | Involved in conjugation |
| 8 | T552_01691 | 1.399000839 | 1.90E-30 | *rgs1* (Sp) | Pheromone response MAPK cascade |
| 8 | T552_03222 | 1.187047091 | 4.11E-22 | *sfc2* (Sp) | Transcription initiation |
| 8 | T552_01719 | 1.132113075 | 2.46E-19 | SPCC1020.07 | Pyrimidine nucleoside salvage |
| 8 | T552_01591 | 1.108511168 | 7.00E-20 | – | Uncharacterized protein |
| 8 | T552_02274 | 1.069895615 | 3.61E-12 | *kha1* (Sp) | Intracellular potassium ion homeostasis |
| 8 | T552_02827 | 0.871860409 | 9.44E-12 | – | Uncharacterized protein |
| 8 | T552_03275 | 0.866236083 | 7.71E-12 | SPAC2H10.01 | DNA-binding transcription factor |
| 8 | T552_02205 | 0.843227814 | 1.52E-10 | *tef3* (Sp) | Cytoplasmic translation |
| 9 | T552_02960 | 6.043734581 | 2.14E-268 | *eng2* (Sp) | Endo-1,3-beta-glucanase |
| 9 | T552_01923 | 5.010302229 | 5.73E-136 | – | Uncharacterized protein |
| 9 | T552_02591 | 3.860423617 | 4.29E-126 | – | Uncharacterized protein |
| 9 | T552_01993 | 3.782894689 | 5.65E-95 | *stg1* (Sp) | Mitotic cytokinesis |
| 9 | T552_03518 | 3.60222119 | 1.07E-49 | *mug113* (Sp) | Meiotic cell cycle |
| 9 | T552_01757 | 3.469389953 | 2.55E-98 | *mde10* (Sp, Af) | Ascospore wall assembly, cell wall integrity |
| 9 | T552_00241 | 3.186654378 | 6.51E-42 | *spo12* (Sp/Sc) | Cell division, chromosome segregation |
| 9 | T552_01429 | 3.057616606 | 1.58E-36 | *slp1* (Sp) | Meiotic phase transition |
| 9 | T552_02590 | 2.958025389 | 8.13E-69 | *trm140* (Sp) | tRNA C3-cytosine methylation |
| 9 | T552_01185 | 2.87518699 | 2.23E-57 | *mid2* (Sp) | Cell division, septum formation |
| 10 | T552_03344 | 3.914918193 | 0.00E+00 | *msa1* (Sp) | Negative regulation of conjugation |
| 10 | T552_02349 | 3.449113926 | 0.00E+00 | – | Uncharacterized protein |
| 10 | T552_03233 | 3.425313953 | 0.00E+00 | – | Uncharacterized protein |
| 10 | T552_03164 | 3.072138162 | 1.11E-300 | *och1* (Sp) | Cell wall mannoprotein biosynthetic process |
| 10 | T552_00770 | 2.581111679 | 5.13E-178 | *hsr1* (Sp) | Response to starvation |
| 10 | T552_00367 | 1.987481232 | 5.31E-109 | *cdr2* (Sp) | Regulation of extent of cell growth |
| 10 | T552_02331 | 1.919802832 | 2.49E-115 | *mei2* (Sp) | Meiotic cell cycle |
| 10 | T552_00539 | 1.822372452 | 3.17E-101 | *dpb3* (Sp) | Protein folding in endoplasmic reticulum |
| 10 | T552_00009 | 1.813629798 | 2.41E-71 | *tos4* (Sp/Sc) | DNA damage response, G1/S phase transition |
| 10 | T552_00715 | 1.608295571 | 2.99E-69 | *bst1* (Sp/Sc) | Protein transport, cell wall integrity |
| 11 | T552_00710 | 4.026186063 | 0.00E+00 | *spc24* (Sp) | Cell division, kinetochore complex, mitosis |
| 11 | T552_01444 | 3.947747021 | 0.00E+00 | – | Uncharacterized protein |
| 11 | T552_02900 | 3.914824191 | 0.00E+00 | *hht1* (Sp) | Heterochromatin formation |
| 11 | T552_01993 | 3.70055861 | 0.00E+00 | *stg1* (Sp) | Mitotic cytokinesis |
| 11 | T552_02650 | 3.56743755 | 0.00E+00 | *nuf2* (Sp) | Cell division, kinetochore complex, mitosis |
| 11 | T552_02752 | 3.538180529 | 0.00E+00 | *spc25* (Sp) | Cell division, kinetochore complex, mitosis |
| 11 | T552_00241 | 3.514638647 | 0.00E+00 | *spo12* (Sp/Sc) | Cell division, chromosome segregation |
| 11 | T552_01429 | 3.469459734 | 0.00E+00 | *slp1* (Sp) | Mitotic phase transition |
| 11 | T552_00303 | 3.436737011 | 0.00E+00 | *mob1* (Sp) | Cell division, mitosis |
| 11 | T552_02095 | 3.245276686 | 0.00E+00 | *hta2* (Sp) | Homologous chromosome segregation |

(*Continued on next page*)

**TABLE 1** Top differentially upregulated genes per cluster (*Continued*)

| Cluster | Feature name | Log2 FC[a] | Adj. *P*-value | Homolog[b] | Function |
|---|---|---|---|---|---|
| 12 | T552_02875 | 7.950214856 | 0.00E+00 | – | Uncharacterized protein |
| 12 | T552_00592 | 7.570781017 | 0.00E+00 | – | Uncharacterized protein |
| 12 | T552_02791 | 5.9273682 | 0.00E+00 | *eng1* (Sp) | Endo-1,3-beta-D-glucosidase |
| 12 | T552_04075 | 5.918664163 | 6.06E-277 | *bgl2* (Sp) | Beta-glucan metabolic process |
| 12 | T552_04055 | 5.660992191 | 0.00E+00 | – | Uncharacterized protein |
| 12 | T552_01051 | 5.556713779 | 0.00E+00 | – | Uncharacterized protein |
| 12 | T552_01340 | 5.469897438 | 6.80E-307 | *gas4/5* (Sp) | Ascospore-type prospore assembly |
| 12 | T552_01817 | 4.757705628 | 0.00E+00 | – | Uncharacterized protein |
| 12 | T552_00288 | 4.753421611 | 0.00E+00 | *pil1* (Sp) | Eisosome assembly |
| 12 | T552_01968 | 4.474771548 | 0.00E+00 | SPAC6G9.14 | Nuclear-transcribed mRNA catabolic process |
| 13 | T552_01984 | 6.862502708 | 1.53E-211 | – | Uncharacterized protein |
| 13 | T552_02979 | 6.284615411 | 2.33E-156 | – | Uncharacterized protein |
| 13 | T552_00122 | 6.030566902 | 2.99E-133 | – | Uncharacterized protein |
| 13 | T552_04075 | 5.558635838 | 1.81E-36 | *bgl2* (Sp) | Cell wall glucan beta-glucosidase |
| 13 | T552_01745 | 5.471704405 | 1.95E-91 | – | Uncharacterized protein |
| 13 | T552_02877 | 5.402237973 | 4.48E-78 | – | Uncharacterized protein |
| 13 | T552_00472 | 5.374174788 | 1.89E-71 | *pmp3* (Sp) | Plasma membrane proteolipid |
| 13 | T552_02662 | 5.315038816 | 1.24E-90 | – | Uncharacterized protein |
| 13 | T552_01444 | 5.209923916 | 1.91E-69 | – | Uncharacterized protein |
| 13 | T552_00853 | 5.065249633 | 1.95E-89 | *cdc13* (Sp) | Control point of mitotic cell cycle |

[a]FC = fold change.
[b]Sp = *S. pombe*; Pc = *P. carinii*; Sc = *C. cerevisiae*; Af = *A. fumigatus*.
[c]"–" denotes no identifiable homolog detected among the surveyed phylogenetically related fungi; it does not indicate missing data.

*adg3* was not downregulated in Cluster C4, suggesting the functions of these two genes may not be universally related.

GO terms were plotted in a line graph across the pseudotime-ordered clusters to illustrate the upregulation of biosynthetic and metabolic processes in the early clusters (Fig. 2B). The graph shows significant upregulation of these pathways in Clusters C1–C4, with a marked peak in early stages, aligning with the high metabolic and translational activity observed in these clusters. As cells progress through pseudotime, there is a decline in the expression of these pathways, reflecting a shift away from biosynthesis and metabolism. A resurgence of metabolic and biosynthetic activity occurs in Cluster C11 (Fig. 2B), just prior to the completion of sexual reproduction.

## Mating-competent trophic clusters (C5, C6, and C8) show activation of pheromone signaling and repression of mitotic activity

Clusters C5, C6, and C8 reflect a transition toward a mating-competent state with activation of mating-related signaling and continued repression of mitotic activity. In Cluster C5 (Table 1), upregulation of *mam2*, *rgs1*, and *spk1* indicates activation of pheromone signaling and mating pathways, likely in response to environmental cues or nutrient limitation. This is supported by increased expression of *fil1* and *adg3*, suggesting nutrient stress and metabolic remodeling. Elevated *tef3* and *sfc2* may aid in the synthesis and trafficking of mating-related proteins. The upregulation of *adg3* (putative β-glucosidase), although not directly linked to mating, occurs alongside mating genes. The enzyme *adg3* breaks down glucans outside the cell and may be involved in scavenging substrates during mating, preparing the cells for later β-glucan synthesis in Clusters C12 and C13. Downregulation of mitotic and chromosome segregation genes (*spc24*, *nuf2*, *spo12*, and *mob1*), chromatin packaging (*hht3*), and cell wall remodeling (*eng1* and *bgl2*) indicates suppression of cell division, suggesting entry into the mating program. The continued downregulation of *eng1* across these clusters implies its specific role in ascus development. GO term enrichment for cell communication and pheromone response, MAPK signaling supports activation of mating-related processes (Table 3).

**TABLE 2** Top differentially downregulated genes per cluster

| Cluster | Feature name | Log2 FC[a] | Adj. *P*-value | Homolog[b] | Function |
|---|---|---|---|---|---|
| 1 | T552_01993 | −2.02302559 | 1.30E-11 | *stg1* (Sp) | Mitotic cytokinesis |
| 1 | T552_00009 | −1.985927158 | 1.37E-15 | *tos4* (Sp) | Negative regulation of transcription |
| 1 | T552_01551 | −1.823680248 | 5.18E-14 | *adg3* (Sp) | Probable secreted beta-glucosidase |
| 1 | T552_01771 | −1.794698037 | 2.61E-09 | –[c] | Uncharacterized protein |
| 1 | T552_00385 | −1.747125052 | 2.08E-17 | *loz1* (Sp) | Negative regulation of transcription |
| 1 | T552_01719 | −1.72008476 | 2.83E-17 | SPCC1020.07 | Pyrimidine nucleoside salvage |
| 1 | T552_02201 | −1.703883244 | 6.19E-12 | *gas1* (Sp) | Beta-D-glucan biosynthetic process |
| 1 | T552_03222 | −1.691671515 | 1.35E-17 | *sfc2* (Sp) | Regulation of transcription |
| 1 | T552_03164 | −1.665785615 | 2.71E-13 | *och1* (Sp) | Cell wall mannoprotein biosynthetic |
| 1 | T552_02875 | −1.626870626 | 3.69E-03 | – | Uncharacterized protein |
| 2 | T552_02875 | −3.263430079 | 1.98E-231 | – | Uncharacterized protein |
| 2 | T552_01444 | −2.646945236 | 6.70E-281 | – | Uncharacterized protein |
| 2 | T552_00710 | −2.499156411 | 0.00E+00 | *spc24* (Sp) | Cell division |
| 2 | T552_01984 | −2.361132595 | 0.00E+00 | – | Uncharacterized protein |
| 2 | T552_02791 | −2.34452429 | 3.04e-312 | *eng1* (Sp) | Endo-1,3-beta-D-glucosidase |
| 2 | T552_01551 | −2.250793221 | 0.00E+00 | *adg3* (Sp) | Probable secreted beta-glucosidase |
| 2 | T552_00241 | −2.062138663 | 1.08E-284 | *spo12* (Sp/Sc) | Cell division, chromosome segregation |
| 2 | T552_02650 | −1.992792701 | 1.62E-235 | *nuf2* (Sp) | Cell division, kinetochore complex, mitosis |
| 2 | T552_02752 | −1.957439468 | 1.37E-302 | *spc25* (Sp) | Cell division, kinetochore complex, mitosis |
| 2 | T552_01429 | −1.931135615 | 2.97E-237 | *slp1* (Sp) | Meiotic phase transition |
| 3 | T552_02875 | −2.381541896 | 3.24E-97 | – | Uncharacterized protein |
| 3 | T552_01993 | −1.918846029 | 3.90E-248 | *stg1* (Sp) | Mitotic cytokinesis |
| 3 | T552_01429 | −1.817958028 | 6.19E-138 | *slp1* (Sp) | Meiotic phase transition |
| 3 | T552_02252 | −1.801795371 | 9.45E-238 | *ace2* (Sp) | Regulation of transcription |
| 3 | T552_00241 | −1.736593471 | 4.70E-139 | *spo12* (Sp/Sc) | Cell division, chromosome segregation |
| 3 | T552_01984 | −1.734938823 | 5.41E-152 | – | Uncharacterized protein |
| 3 | T552_01551 | −1.713567568 | 3.04E-253 | *adg3* (Sp) | Probable secreted beta-glucosidase |
| 3 | T552_02900 | −1.702899088 | 1.78E-254 | *hht1* (Sp) | Heterochromatin formation |
| 3 | T552_01444 | −1.630038717 | 6.98E-94 | – | Uncharacterized protein |
| 3 | T552_02791 | −1.599102083 | 4.04E-115 | *eng1* (Sp) | Endo-1,3-beta-D-glucosidase |
| 4 | T552_02875 | −2.265205412 | 1.14E-36 | – | Uncharacterized protein |
| 4 | T552_01444 | −1.71772971 | 1.66E-40 | – | Uncharacterized protein |
| 4 | T552_00241 | −1.360535946 | 3.15E-38 | *spo12* (Sp) | Cell division |
| 4 | T552_02900 | −1.31245943 | 1.38E-67 | *hht1* (Sp) | Heterochromatin formation |
| 4 | T552_00710 | −1.291310802 | 2.45E-41 | *spc24* (Sp) | Cell division, kinetochore complex, mitosis |
| 4 | T552_04055 | −1.147681865 | 4.72E-18 | – | Uncharacterized protein |
| 4 | T552_02752 | −1.096036409 | 2.88E-31 | *spc25* (Sp) | Cell division, kinetochore complex, mitosis |
| 4 | T552_02791 | −1.06615682 | 1.14E-23 | *eng1* (Sp) | Endo-1,3-beta-D-glucosidase |
| 4 | T552_01422 | −1.041119693 | 3.53E-23 | – | Uncharacterized protein |
| 4 | T552_01661 | −0.955375964 | 4.10E-22 | *hrk1* (Sp/Sc) | Mitotic cell cycle, salt stress response |
| 5 | T552_02875 | −2.609718812 | 2.95E-35 | – | Uncharacterized protein |
| 5 | T552_00710 | −1.669866139 | 4.59E-48 | *spc24* (Sp) | Cell division, kinetochore complex, mitosis |
| 5 | T552_02900 | −1.646578146 | 1.19E-77 | *hht3* (Sp) | Heterochromatin formation |
| 5 | T552_02791 | −1.426229969 | 1.56E-30 | *eng1* (Sp) | Endo-1,3-beta-D-glucosidase |
| 5 | T552_01444 | −1.423341617 | 1.80E-23 | – | Uncharacterized protein |
| 5 | T552_00241 | −1.354376961 | 3.68E-29 | *spo12* (Sp/Sc) | Cell division, chromosome segregation |
| 5 | T552_01984 | −1.301992311 | 9.01E-30 | – | Uncharacterized protein |
| 5 | T552_04075 | −1.25399417 | 5.21E-10 | *bgl2* (Sp) | Cell wall glucan beta-glucosidase |
| 5 | T552_02650 | −1.251616759 | 2.06E-24 | *nuf2* (Sp) | Cell division, kinetochore complex, mitosis |
| 5 | T552_00303 | −1.142279837 | 2.55E-25 | *mob1* (Sp) | Cell division |
| 6 | T552_02875 | −2.635703145 | 7.12E-21 | – | Uncharacterized protein |
| 6 | T552_01444 | −2.067839391 | 2.45E-24 | – | Uncharacterized protein |

*(Continued on next page)*

**TABLE 2** Top differentially downregulated genes per cluster (*Continued*)

| Cluster | Feature name | Log2 FC[a] | Adj. *P*-value | Homolog[b] | Function |
|---|---|---|---|---|---|
| 6 | T552_02791 | −1.90446052 | 2.64E-29 | *eng1* (Sp) | Endo-1,3-beta-D-glucosidase |
| 6 | T552_04075 | −1.880304682 | 3.63E-11 | *bgl2* (Sp) | Glucan 1,3-beta-glucosidase |
| 6 | T552_00710 | −1.787230068 | 2.81E-32 | *spc24* (Sp) | Cell division |
| 6 | T552_01984 | −1.769330512 | 8.33E-31 | – | Uncharacterized protein |
| 6 | T552_01429 | −1.624612895 | 1.38E-21 | *slp1* (Sp) | Meiotic phase transition |
| 6 | T552_02752 | −1.609252261 | 3.37E-27 | *spc25* (Sp) | Cell division |
| 6 | T552_01993 | −1.605778905 | 2.81E-32 | *stg1* (Sp) | Mitotic cytokinesis |
| 6 | T552_02650 | −1.496521179 | 1.27E-19 | *nuf2* (Sp) | Cell division |
| 7 | T552_02875 | −2.088997784 | 5.30E-08 | – | Uncharacterized protein |
| 7 | T552_00472 | −1.526959866 | 2.61E-09 | *pmp3* (Sp) | Plasma membrane proteolipid |
| 7 | T552_01444 | −1.452641697 | 2.71E-08 | – | Uncharacterized protein |
| 7 | T552_00303 | −1.439905709 | 1.60E-11 | *mob1* (Sp) | Cell division, mitotic cytokinesis |
| 7 | T552_01422 | −1.431380192 | 9.65E-10 | – | Uncharacterized protein |
| 7 | T552_02752 | −1.370179537 | 4.36E-11 | *spc25* (Sp) | Cell division, kinetochore complex, mitosis |
| 7 | T552_03389 | −1.302519519 | 4.06E-13 | *cdc13* (Sp) | Control point of mitotic cell cycle |
| 7 | T552_02095 | −1.241831872 | 1.73E-16 | *hta2* (Sp) | Heterochromatin formation |
| 7 | T552_02282 | −1.239974236 | 1.03E-10 | *klp2* (Sp/Sc) | Karyogamy involved in conjugation |
| 7 | T552_00710 | −1.192416259 | 3.10E-09 | *spc24* (Sp) | Cell division, kinetochore complex, mitosis |
| 8 | T552_02875 | −2.63978455 | 5.62E-06 | – | Uncharacterized protein |
| 8 | T552_01429 | −1.733525059 | 3.91E-07 | *slp1* (Sp) | Meiotic phase transition |
| 8 | T552_00241 | −1.678041526 | 2.08E-07 | *spo12* (Sp) | Cell division, chromosome segregation |
| 8 | T552_02650 | −1.606856448 | 1.47E-06 | *nuf2* (Sp) | Cell division, kinetochore complex, mitosis |
| 8 | T552_00710 | −1.600357573 | 3.79E-08 | *spc24* (Sp) | Cell division, kinetochore complex, mitosis |
| 8 | T552_02900 | −1.586486317 | 1.37E-13 | *hht1* (Sp) | Heterochromatin formation |
| 8 | T552_01422 | −1.508555043 | 5.20E-06 | – | Uncharacterized protein |
| 8 | T552_02752 | −1.415249518 | 1.12E-06 | *spc25* (Sp) | Cell division, kinetochore complex, mitosis |
| 8 | T552_04188 | −1.354565642 | 1.04E-17 | – | Uncharacterized protein |
| 8 | T552_02252 | −1.293465293 | 2.23E-07 | *ace2* (Sp) | Regulation of transcription |
| 9 | T552_02875 | −2.720841643 | 6.89E-02 | – | Uncharacterized protein |
| 9 | T552_02791 | −2.321388252 | 1.62E-03 | *eng1* (Sp) | Endo-1,3-beta-D-glucosidase |
| 9 | T552_01551 | −1.996957532 | 1.04E-05 | *adg3* (Sp) | Probable secreted beta-glucosidase adg3 |
| 9 | T552_01340 | −1.630037384 | 1.29E-01 | *gas4/5* (Sp) | Ascospore wall beta-glucan |
| 9 | T552_02201 | −1.482413155 | 1.24E-03 | *gas1* (Sp) | 1,3-beta-glucanosyltransferase |
| 9 | T552_00644 | −1.463013305 | 2.92E-03 | – | Uncharacterized protein |
| 9 | T552_02349 | −1.298004055 | 4.36E-03 | – | Uncharacterized protein |
| 9 | T552_00243 | −1.218413555 | 4.63E-04 | *msh6* (Sp) | Mismatch repair |
| 9 | T552_02092 | −1.186786543 | 3.18E-04 | *gpc1* (Sp) | Phosphatidylcholine biosynthetic process |
| 9 | T552_00457 | −1.182357372 | 1.03E-03 | *suc1* (Sp) | Cell division |
| 10 | T552_01551 | −3.64467115 | 5.19E-78 | *adg3* (Sp) | Probable secreted beta-glucosidase |
| 10 | T552_00710 | −1.916083523 | 6.85E-18 | *spc24* (Sp) | Cell division, kinetochore complex, mitosis |
| 10 | T552_02900 | −1.887653949 | 1.14E-29 | *hht1* (Sp) | Heterochromatin formation |
| 10 | T552_01429 | −1.863403271 | 2.59E-13 | *slp1* (Sp) | Cell division |
| 10 | T552_02252 | −1.733836658 | 2.08E-19 | *ace2* (Sp) | Regulation of transcription for late mitotic events |
| 10 | T552_01444 | −1.70191325 | 1.30E-09 | – | Uncharacterized protein |
| 10 | T552_02875 | −1.642532679 | 3.11E-05 | – | Uncharacterized protein |
| 10 | T552_02142 | −1.501430929 | 6.25E-26 | *spk1* (Sp) | Pheromone response MAPK cascade |
| 10 | T552_04075 | −1.439055591 | 3.15E-04 | *bgl2* (Sp) | Cell wall glucan beta-glucosidase |
| 10 | T552_00241 | −1.434163815 | 5.62E-10 | *spo12* (Sp) | Cell division, chromosome segregation |
| 11 | T552_02826 | −2.526339795 | 3.43E-19 | – | Uncharacterized protein |
| 11 | T552_01551 | −2.351488921 | 2.55E-141 | *adg3* (Sp) | Probable secreted beta-glucosidase |
| 11 | T552_02426 | −1.002534947 | 2.64E-50 | *krp1* (Sp) | Peptide mating pheromone maturation |
| 11 | T552_01691 | −0.938984244 | 1.60E-42 | *rgs1* (Sp) | Pheromone response MAPK cascade |

(*Continued on next page*)

**TABLE 2** Top differentially downregulated genes per cluster (*Continued*)

| Cluster | Feature name | Log2 FC$^a$ | Adj. *P*-value | Homolog$^b$ | Function |
|---|---|---|---|---|---|
| 11 | T552_02233 | −0.92826009 | 6.07E-34 | *mam2* (Sp) | Pheromone-dependent signal transduction |
| 11 | T552_01937 | −0.909771914 | 2.75E-48 | *ole1* (Sp) | Unsaturated fatty acid biosynthetic process |
| 11 | T552_01110 | −0.771051131 | 1.28E-39 | – | Uncharacterized protein |
| 11 | T552_01113 | −0.715019498 | 3.66E-35 | *krp1* (Sp) | Peptide mating pheromone maturation |
| 11 | T552_01049 | −0.714258969 | 4.85E-28 | *fil1* (Sp) | Positive regulation of autophagy |
| 11 | T552_03467 | −0.699824464 | 8.52E-30 | *pi067* (Sp) | Cellular response to phosphate starvation |
| 12 | T552_01551 | −2.214667571 | 1.11E-24 | *adg3* (Sp) | Probable secreted beta-glucosidase |
| 12 | T552_01841 | −2.206088588 | 1.30E-36 | *ptr2* (Sp) | Protein transport |
| 12 | T552_03182 | −2.19448151 | 3.37E-38 | – | Uncharacterized protein |
| 12 | T552_02142 | −2.094744671 | 1.01E-27 | *spk1* (Sp) | Pheromone response MAPK cascade |
| 12 | T552_01540 | −2.03538371 | 5.88E-39 | *rps29* (Sp) | Cytoplasmic translation |
| 12 | T552_04148 | −2.012992802 | 7.69E-35 | SPAC1F5.02 | Protein folding |
| 12 | T552_00052 | −2.008201244 | 1.28E-34 | *rpl1101* (Sp) | Cytoplasmic translation |
| 12 | T552_00654 | −2.004708689 | 5.53E-37 | *wos2* (Sp) | Protein folding |
| 12 | T552_02765 | −1.981660307 | 5.85E-41 | *ppi1* (Sp) | Protein folding |
| 12 | T552_00336 | −1.963773562 | 4.07E-35 | *rps2801* (Sp) | Cytoplasmic translation |
| 13 | T552_02608 | −5.708286942 | 3.21E-11 | SPAC589.06c | Protein targeting to ER |
| 13 | T552_03174 | −5.659593019 | 3.15E-11 | *nrs1* (Sp/Sc) | Cytoplasmic translation |
| 13 | T552_02212 | −5.287889921 | 1.40E-13 | *grx2* (Sp) | Cell redox homeostasis |
| 13 | T552_02709 | −5.057122467 | 3.04E-11 | *rpl2502* (Sp) | Cytoplasmic translation |
| 13 | T552_02357 | −5.051945566 | 2.94E-11 | *epl1* (Sp) | DNA repair, chromatin remodeling |
| 13 | T552_01490 | −5.000560057 | 4.26E-11 | *srp2* (Sp/Sc) | RNA splicing, targeting proteins to the ER |
| 13 | T552_02666 | −4.934666772 | 1.84E-11 | *rps102* (Sp) | Cytoplasmic translation |
| 13 | T552_02385 | −4.927660702 | 7.99E-13 | *rps2201* (Sp) | Cytoplasmic translation |
| 13 | T552_03222 | −4.904168018 | 3.70E-09 | *sfc2* (Sp/Sc) | Regulation of transcription |
| 13 | T552_01210 | −4.898617598 | 6.23E-12 | *mdm28* (Sp) | Mitochondrial translation |

$^a$FC = fold change.
$^b$Sp = *S. pombe*; Pc = *P. carinii*; Sc = *C. cerevisiae*; Af = *A. fumigatus*.
$^c$"–" denotes no identifiable homolog detected among the surveyed phylogenetically related fungi; it does not indicate missing data.

In Cluster C6, upregulation of multiple transcripts for pheromone maturation peptidases such as *iph1* and *krp1* (Table 1) is observed, while mitotic and cell cycle regulators remain repressed (Table 2). Downregulation of *bgl2* suggests that *adg3* activity is distinct from the regulated endo-1,3-β-D-glucosidase activity of *eng1* and *bgl2*. GO term enrichment for pheromone response and signal transduction reinforces this cluster's focus on mating (Table 3).

Cluster C8 shows upregulation of *mam2* and *rgs1* (Table 1), indicating pheromone response pathway activation. Concurrent expression of *adg3*, ion transporters (*kha1*), and trafficking-related genes (*sfc2*, *SPCC1020.07*, *SPAC2H10.01*) suggests resource reallocation for mating, possibly including membrane remodeling. Downregulation of core cell cycle and mitosis regulators (*slp1*, *spo12*, *nuf2*, *spc24*, *spc25*, *hht1*, and *ace2*) indicates suppression of mitotic progression and chromosomal segregation, consistent with a G1 arrest or early meiotic entry. These transcriptional changes indicate a halt in proliferation to prioritize mating readiness. GO term enrichment for cell communication and pheromone response, MAPK signaling further supports mating activation (Table 3).

GO terms related to signaling, mating, and conjugation pathways were plotted across the pseudotime-ordered clusters to capture the dynamic regulation in Cluster C5, C6, and C8 (Fig. 2C). A clear increase in these pathways is observed from Cluster C5 to Cluster C6, followed by another peak in Cluster C8, indicating the activation of mating processes in these clusters. This pattern aligns with the upregulation of mating-related genes (*mam2* and *rgs1*) (Table 1), reinforcing the activation of mating signaling as the cells transition through these stages.

**TABLE 3** Top 5 GO terms per cluster

| Cluster | GO term name | GO term ID | Enrichment score | Adjusted *P*-value |
|---------|--------------|------------|------------------|--------------------|
| 1 | Structural constituent of ribosome | GO:0003735 | 39.56 | 2.73E-40 |
| 1 | Metabolic process | GO:0008152 | 4.83 | 2.96E-25 |
| 1 | Protein metabolic process | GO:0019538 | 2.18 | 3.26E-29 |
| 1 | Biosynthetic process | GO:0009058 | 2.13 | 2.11E-31 |
| 1 | Sexual reproduction | GO:0019953 | 1.71 | 1.51E-08 |
| 2 | Binding | GO:0005488 | 9.12 | 1.77E-19 |
| 2 | Molecular function | GO:0003674 | 8.45 | 2.05E-30 |
| 2 | Biological process | GO:0008150 | 7.98 | 1.02E-29 |
| 2 | Protein binding | GO:0005515 | 5.52 | 5.05E-23 |
| 2 | Metabolic process | GO:0008152 | 4.54 | 2.96E-25 |
| 3 | Cytoplasmic translation | GO:0002181 | 47.7 | 2.01E-48 |
| 3 | Peptide biosynthetic process | GO:0043043 | 41.33 | 4.66E-42 |
| 3 | Biosynthetic process | GO:0009058 | 30.68 | 2.11E-31 |
| 3 | Metabolic process | GO:0008152 | 24.53 | 2.96E-25 |
| 3 | Ribosome biogenesis | GO:0042254 | 8.22 | 6.07E-09 |
| 4 | Cytoplasmic translation | GO:0002181 | 23.2 | 2.01E-48 |
| 4 | Protein metabolic process | GO:0019538 | 16.88 | 3.26E-29 |
| 4 | Biosynthetic process | GO:0009058 | 7.81 | 2.11E-31 |
| 4 | Metabolic process | GO:0008152 | 4.36 | 2.96E-25 |
| 4 | Ribosome biogenesis | GO:0042254 | 2.53 | 6.07E-09 |
| 5 | Chaperone-mediated protein folding | GO:0061077 | 5.59 | 2.59E-06 |
| 5 | Protein folding | GO:0006457 | 4.6 | 2.49E-05 |
| 5 | Conjugation with cellular fusion | GO:0000747 | 2.53 | 2.22E-04 |
| 5 | Cell communication | GO:0007154 | 2.06 | 1.73E-12 |
| 5 | Cellular response to stimulus | GO:0051716 | 1.33 | 1.92E-13 |
| 6 | Regulation of conjugation with cellular fusion | GO:0031137 | 3.41 | 3.89E-04 |
| 6 | Pheromone response MAPK cascade | GO:0071507 | 3.33 | 1.43E-02 |
| 6 | Intracellular signal transduction | GO:0035556 | 2.65 | 6.75E-13 |
| 6 | Cell communication | GO:0007154 | 2.45 | 1.73E-12 |
| 6 | Conjugation with cellular fusion | GO:0000747 | 2.19 | 2.22E-04 |
| 7 | Molecular function | GO:0003674 | 4.8 | 2.05E-30 |
| 7 | RNA polymerase II sequence-specific DNA binding | GO:0000978 | 4.65 | 2.22E-05 |
| 7 | DNA-binding transcription factor | GO:0000981 | 4.58 | 2.61E-05 |
| 7 | Cis-regulatory region sequence-specific DNA binding | GO:0000987 | 4.51 | 3.06E-05 |
| 7 | Negative regulation of transcription | GO:0000122 | 1.3 | 6.93E-03 |
| 8 | DNA-binding transcription repressor activity | GO:0001227 | 2.81 | 1.54E-03 |
| 8 | Cell communication | GO:0007154 | 2.45 | 1.73E-12 |
| 8 | Conjugation with cellular fusion | GO:0000747 | 2.19 | 2.22E-04 |
| 8 | Pheromone response MAPK cascade | GO:0071507 | 1.85 | 1.43E-02 |
| 8 | Cellular response to stimulus | GO:0051716 | 1.34 | 1.92E-13 |
| 9 | Molecular function | GO:0003674 | 9.64 | 2.05E-30 |
| 9 | Biological process | GO:0008150 | 9.26 | 1.02E-29 |
| 9 | Negative regulation of cellular process | GO:0048523 | 5.44 | 1.71E-10 |
| 9 | Negative regulation of biological process | GO:0048519 | 5.22 | 3.92E-10 |
| 9 | Cell communication | GO:0007154 | 2.2 | 1.73E-12 |
| 9 | Cellular response to stimulus | GO:0051716 | 0.98 | 1.92E-13 |
| 10 | Meiotic cell cycle | GO:0051321 | 11.21 | 2.68E-06 |
| 10 | Metabolic process | GO:0008152 | 5.35 | 2.96E-25 |
| 10 | Cellular response to stimulus | GO:0051716 | 5.06 | 1.92E-13 |
| 10 | Sexual reproduction | GO:0019953 | 5.04 | 1.51E-08 |
| 10 | Cell communication | GO:0007154 | 3.81 | 1.73E-12 |

(*Continued on next page*)

**TABLE 3** Top 5 GO terms per cluster (*Continued*)

| Cluster | GO term name | GO term ID | Enrichment score | Adjusted *P*-value |
|---|---|---|---|---|
| 11 | Mitotic cell cycle | GO:0000278 | 29.39 | 4.09E-30 |
| 11 | Cell cycle | GO:0007049 | 25.38 | 4.19E-26 |
| 11 | Cellular response to stimulus | GO:0051716 | 12.72 | 1.92E-13 |
| 11 | Nuclear division | GO:0000280 | 10.23 | 5.85E-11 |
| 11 | Chromosome segregation | GO:0007059 | 9.54 | 2.85E-10 |
| 12 | Cell cycle | GO:0007049 | 25.07 | 4.19E-26 |
| 12 | Nuclear division | GO:0000280 | 9.69 | 5.85E-11 |
| 12 | Chromosome segregation | GO:0007059 | 7.68 | 2.85E-10 |
| 12 | Metabolic process | GO:0008152 | 6.42 | 2.96E-25 |
| 12 | Mitotic cell cycle | GO:0051321 | 5.45 | 2.68E-06 |
| 13 | Mitotic cell cycle | GO:0000278 | 24.42 | 4.09E-30 |
| 13 | Cell communication | GO:0007154 | 11.76 | 1.73E-12 |
| 13 | Cellular response to stimulus | GO:0051716 | 11.47 | 1.92E-13 |
| 13 | Sexual reproduction | GO:0019953 | 6.69 | 1.51E-08 |
| 13 | Meiotic cell cycle | GO:0051321 | 5.45 | 2.68E-06 |

## Cluster C7 functions as a transcriptional checkpoint separating mating and meiosis before sexual reproduction

Cluster C7 exhibits a unique transcriptional profile that marks a regulatory pause between mating and sexual reproduction. While genes involved in transcriptional regulation, such as *fep1* (iron uptake regulator), are upregulated (Table 1), no mating or meiotic markers are expressed in this cluster. The downregulation of genes related to cell division and mitotic regulation (*mob1*, *spc25*, *cdc13*, and *spc24*) (Table 2) suggests that Cluster C7 functions as a pause point, halting the progression from mating to sexual reproduction. GO term enrichment (Table 3) reveals terms associated with transcriptional regulation and negative regulation of transcription, further supporting the role of Cluster C7 in this regulatory pause. These findings are also reflected in the line graphs, which show the downregulation of metabolic, biosynthetic, mating signaling, and conjugation processes in Clusters 1–6, confirming the regulatory pause in Cluster C7 (Fig. 2B and C). This shift in gene expression highlights the paused state in Cluster C7 before the cells proceed to sexual reproduction in subsequent clusters.

## Late sexual development (C9–C13) shows progression through meiosis, chromosome segregation, and ascus wall formation

Cluster C9-13 represent the progression of sexual reproduction, with distinct gene expression patterns reflecting the transition from meiotic initiation to ascus formation. In Cluster C9, significant upregulation of meiotic genes (*stg1*, *mug113*, *spo12*, *slp1*, *mde10*, and *mid2*) (Table 1) suggests entry into meiosis, including chromosome segregation, spore wall biosynthesis, and cytokinesis. Upregulation of *eng2* and *mde10*, involved in cell wall β-glucan remodeling, marks the onset of β-glucan metabolism; however, key genes for β-glucan biosynthesis remain absent, with crucial late-stage genes still downregulated. Downregulation of mitotic cell wall enzymes (*eng1*, *gas1*, and *gas4/5*), DNA repair factors (*msh6*), and cell cycle regulators (*suc1* and *gpc1*) indicates suppression of the vegetative cell cycle and mitotic checkpoints (Table 2), allowing a shift to meiosis. GO terms related to negative regulation of molecular function, cellular processes, and response to stimuli reflect coordination for sexual reproduction (Table 3).

Cluster C10 continues meiosis with upregulation of *mei2*, *msa1*, and *tos4*, indicating meiotic initiation and regulation of gene expression. Upregulation of *cdr2*, *och1*, and *bst1* supports cell cycle remodeling, possibly to reorient polarity (Table 1). Downregulation of mitotic and chromatin regulators (*spc24*, *slp1*, *hht1*, and *ace2*) (Table 2) confirms repression of mitosis and chromosome segregation, emphasizing meiotic commitment.

GO term enrichment supports meiosis and sexual reproduction as cells progress toward ascus formation (Table 3).

Cluster C11 shows upregulation of kinetochore genes (*spc24*, *nuf2*, and *spc25*), histone genes (*hht1* and *hta2*), and mitotic exit regulators (*slp1*, *mob1*, and *spo12*), indicating a shift toward robust mitotic proliferation after meiosis (Table 1). Downregulation of meiotic and mating-related genes (*krp1*, *rgs1*, and *mam2*) (Table 2) supports the transition from meiosis to mitosis. GO term enrichment confirms this mitotic phase, marking the only mitosis observed in the life cycle. While downregulation of mating-related genes may be preventing mating inside the ascus.

Cluster C12 reflects a physiological state focused on cell wall remodeling and environmental adaptation rather than proliferation. Upregulation of *eng1*, *bgl2*, *gas4/5*, and *pil1* indicates β-glucan cell wall remodeling and membrane reorganization (Table 1). Concurrent downregulation of ribosomal genes (*rps29*, *rps2801*, and *rpl1101*), protein-folding machinery (*wos2* and *ppi1*), and nutrient acquisition genes (*ptr2*) indicates suppressed growth and signaling pathways. This transcriptional program suggests a differentiated, non-proliferative state, consistent with ascus maturation. GO term enrichment for chromosome segregation and mitotic division supports this conclusion (Table 3).

Cluster C13 shows upregulation of *bgl2* and *pmp3*, indicating cell wall and membrane preservation under stress, and possibly late-stage cell cycle arrest or quiescence in preparation for spore release. *Cdc13*, a telomere-capping protein, is upregulated, supporting genomic stability over replication. Downregulation of genes involved in redox homeostasis (*grx2*), nutrient-responsive cell cycle regulation (*nrs1*), and ribosomal proteins (*rps102*, *rps2201*, and *rpl2502*) (Table 2) suggests suppression of translation and cell growth. Downregulation of *sfc2* and *mdm28* indicates reduced metabolic activity and organelle biogenesis. This gene expression pattern supports the hypothesis that *P. carinii* cells enter a quiescent, structurally stabilized state, likely a dormancy-like response to host stress or nutrient limitation. GO term enrichment highlights the final steps of sexual reproduction and ascus stress response (Table 3).

GO terms associated with sexual reproduction are illustrated in a line graph (Fig. 2D), demonstrating the dynamic upregulation of meiosis and mitosis across the ascus-related Clusters C10-13. A distinct peak in expression around Cluster C12 and C13 marks the culmination of ascus formation. Similarly, GO terms associated with meiosis and mitosis checkpoint signaling exhibit sharp peaks between Cluster C10 and C13, confirming the regulation of meiotic and mitotic divisions across these stages (Fig. 2E). The previously mentioned resurgence of biosynthetic and metabolic pathways (Fig. 2B) tapers down to baseline in Cluster C7.

## Late-stage marker genes are specifically expressed in ascus-forming cells of *P. carinii*

To validate stage-specific gene expression patterns identified by scRNA-seq, reverse transcription quantitative PCR (RT-qPCR) was performed on *P. carinii* RNA isolated from rats treated with anidulafungin, a β-1,3-D-glucan synthase inhibitor that halts ascus production. The inhibition of ascus formation was expected to reduce or eliminate the expression of late-stage ascus markers. RNA from treated animals was compared to RNA from untreated controls containing all life cycle stages. Expression of three scRNA-seq-defined late-stage marker genes, T552_01968 (*mcp2*), T552_01043 (*dmc1*), and T552_01932 (uncharacterized), was measured (Fig. 3A). The late-stage markers were chosen because genes in clusters C1–C8 have more shared functions, and the marker genes for C11–C13 are more specific to these clusters, making them ideal for validating stages.

All three ascus-associated marker genes, *T552_01968* (*mcp1*), *T552_01043* (*dmc1*), and *T552_01932*, were significantly downregulated in the ascus-depleted population compared to untreated controls (Fig. 3B), with relative quantities reduced to 66%, 41%, and 49% of control levels, respectively. These results confirm that expression of these

(A)

(B)

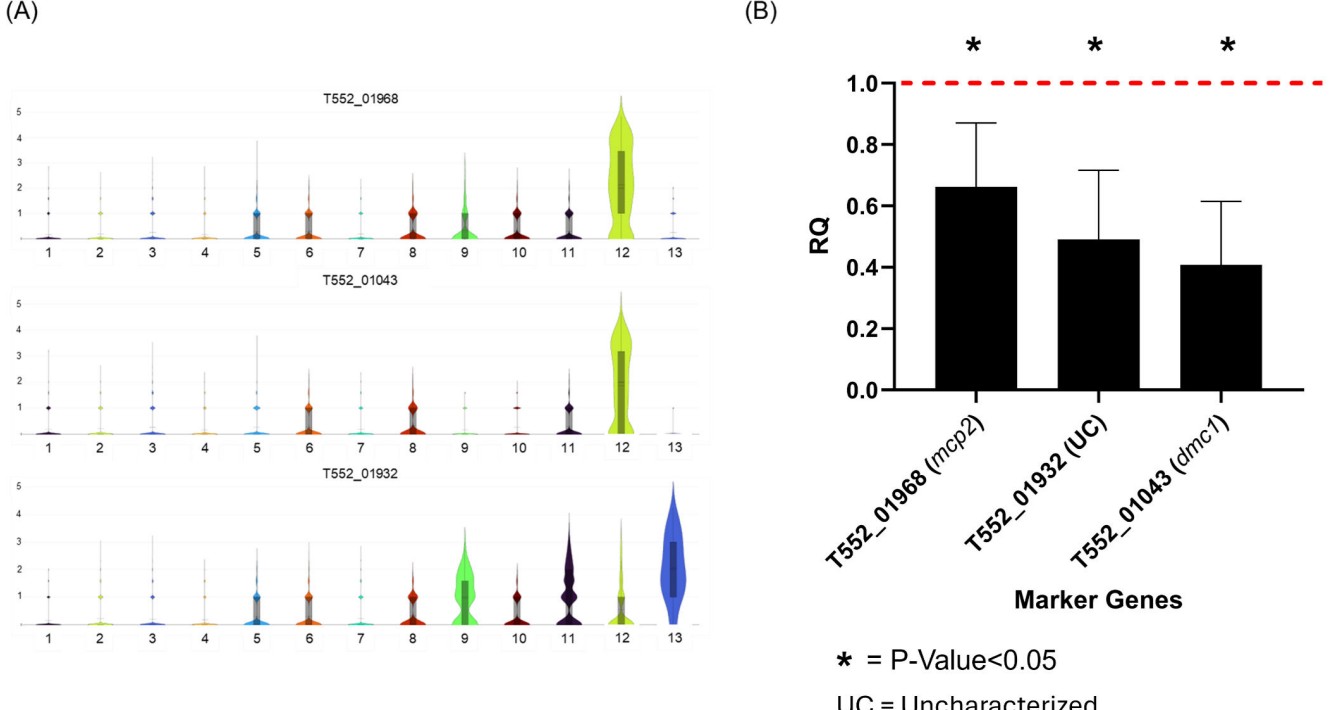

* = P-Value<0.05

UC = Uncharacterized

**FIG 3** Validation of scRNA-seq-defined ascus marker gene expression by RT-qPCR. (A) Violin plots show single-cell expression patterns of three ascus-enriched marker genes T552_01968 (*mcp1*), T552_01932 (uncharacterized), and T552_01043 (*dmc1*) across 13 transcriptional clusters (C1–C13) identified by scRNA-seq. All three genes are strongly expressed in Clusters C12 and C13, consistent with their assignment as late-stage ascus markers. (B) Bar graphs show RT-qPCR validation of these genes using RNA extracted from *P. carinii*-infected rat lungs. Samples were collected from untreated animals (all life cycle stages present, confirmed microscopically) and from animals treated with anidulafungin for 3 weeks (ascus-depleted, confirmed microscopically). Gene expression was normalized to the geometric mean of three internal reference genes: T552_03081 (*TS*), T552_04188 (uncharacterized), and *T552_00947* (uncharacterized). The two novel reference genes were selected from the scRNA-seq data set based on high expression, uniform detection across clusters, and minimal variance ($P \approx 1$). Relative quantities (RQ) were calculated using the $2^{-\Delta\Delta Ct}$ method with primer efficiency correction. All three ascus marker genes showed significant downregulation in the ascus-depleted group (FDR-adjusted $P < 0.01$; $n = 3$). Error bars represent the standard error of the mean (SEM). The red dashed line indicates normalized baseline expression in untreated animals.

late-stage genes is enriched in mature asci and largely absent from earlier developmental forms. These reductions align with the scRNA-seq data, where expression of all three genes is mostly restricted to Clusters C12 and C13 (Fig. 3A), which represent the predicted asci cell types. These data support their roles as late-stage ascus markers, which are transcriptionally reduced in ascus-depleted populations following anidulafungin treatment, consistent with the loss of mature asci in these samples.

These results independently corroborate the scRNA-seq-defined late-stage ascus markers, demonstrating that transcripts enriched in late transcriptional clusters are selectively diminished following pharmacologic blockade of ascus formation. The combined single-cell and qPCR data support a developmental model in which *P. carinii* progresses through discrete transcriptional states that culminate in ascus maturation. Together, these findings establish a transcriptional map for dissecting *in vivo* developmental progression in *P. carinii*.

## DISCUSSION

This study provides the first single-cell transcriptional map of *P. carinii*, revealing a coordinated life cycle progression from early trophic growth to mating, meiosis, and ascus formation. Using scRNA-seq, we identified 13 transcriptionally distinct clusters, which align with morphologically inferred stages, offering a comprehensive model for *P. carinii* life cycle progression in its host-dependent environment (Fig. 4). Our findings

challenge previous models that proposed asexual replication of trophic forms (19), as mitosis is repressed in all trophic clusters (C1–C8), suggesting obligate sexual reproduction. The early trophic clusters (C1–C4, C7) are enriched for metabolic genes, underscoring their focus on growth and nutrient acquisition rather than replication, consistent with observations in other fungi such as *Candida albicans* and *Cryptococcus neoformans* (20–22). In the mating-associated clusters (C5–C8), upregulation of *mam2*, the pheromone receptor, marks the transition to sexual differentiation (23), with dynamic regulation of *mam2* and *map3* differing from previous genomic studies, which anticipated concomitant expression of mating genes. In addition, *adg3*, a β-glucosidase upregulated during mating, suggests its role in cell surface remodeling and nutrient acquisition by β-glucan scavenging, separate from other β-glucan genes that are downregulated (Fig. 5A) (24–26).

Clusters C9 through C13 represent transcriptional states associated with sexual reproduction, including meiosis, mitosis, and sporulation. Expression of T552_02202 (*gsc1*), the catalytic subunit of the β1,3-glucan synthase complex, is suppressed in Clusters C1–C10 and becomes upregulated in Clusters C11–C13, coinciding with the onset of ascus wall formation. In Clusters C12 and C13, we also observed coordinated upregulation of additional β-glucan biosynthesis genes, including members of the *gas*, *bgl*, and *bgs* families. This pattern aligns with ultrastructural evidence that β-glucan is deposited late in development (13). Notably, *adg3* is upregulated earlier in Clusters C5, C6, and C8, suggesting a preparatory role in mobilizing glucan precursors. The tight temporal separation between *adg3*-mediated scavenging and *gsc1*-driven synthesis supports a regulated transition from substrate acquisition to spore wall assembly during the final stages of the sexual cycle (Fig. 5A). Expression of *pil1* was tightly restricted to Cluster C12 in *P. carinii* (Fig. 5B), consistent with its potential role in ascus maturation. In other filamentous fungi such as *Aspergillus nidulans* and *Neurospora crassa*, *pil1* is essential for membrane bending and spore encapsulation during the final stages of sporulation. The co-expression of *pil1* with other sporulation genes in Clusters C12 and C13 supports a conserved function for *pil1* in membrane remodeling and ascus completion in *P. carinii* (27–29). High expression of *eng1* in Cluster C12, a late-stage population enriched for ascus-associated genes, suggests a role in septum breakdown and spore release in *P. carinii*. In *S. pombe* and *S. cerevisiae*, *eng1* and *eng2* are functional paralogs that encode endo-1,3-β-glucanases with conserved GH81 domains (30, 31). Eng1 is required for dissolving the ascus septum during spore release, while *eng2* functions earlier in development. In *P. carinii*, *eng2* is expressed in Cluster C9, consistent with stage-specific expression. This conserved pattern supports a model in which *eng2* acts during early cell wall remodeling and *eng1* functions at the final stage to facilitate spore escape from mature asci. These findings reflect the evolutionary divergence of *P. carinii* from related fungi, likely driven by its host-obligate life cycle. Despite these advancements, the potential underrepresentation of adherent trophic forms (13) lost during BALF collection remains a limitation, and future studies should optimize sampling methods to capture these forms, as they likely exhibit distinct transcriptional profiles.

In comparison with previously hypothesized life cycles, our results support key aspects of traditional models, including the dominance of trophic forms and obligate sexual reproduction leading to ascus formation, which is consistent with early electron microscopy studies. However, we provide novel insights that challenge earlier models. The absence of asexual replication in trophic forms, supported by the lack of mitotic gene expression, contrasts with the notion of binary fission proposed by earlier studies (32). Our findings also align with genomic studies suggesting primary homothallism, as all cells possess a single mating type locus, differing from earlier assumptions of heterothallism (10, 33). Furthermore, the identification of a transcriptional checkpoint in Cluster C7, a regulatory pause between mating and meiosis, has not been previously described. The sequential and distinct transcriptional states we identify further challenge the notion of overlapping processes suggested in earlier models (34, 35), providing a more refined understanding of the progression of the life cycle. Finally, the upregulation

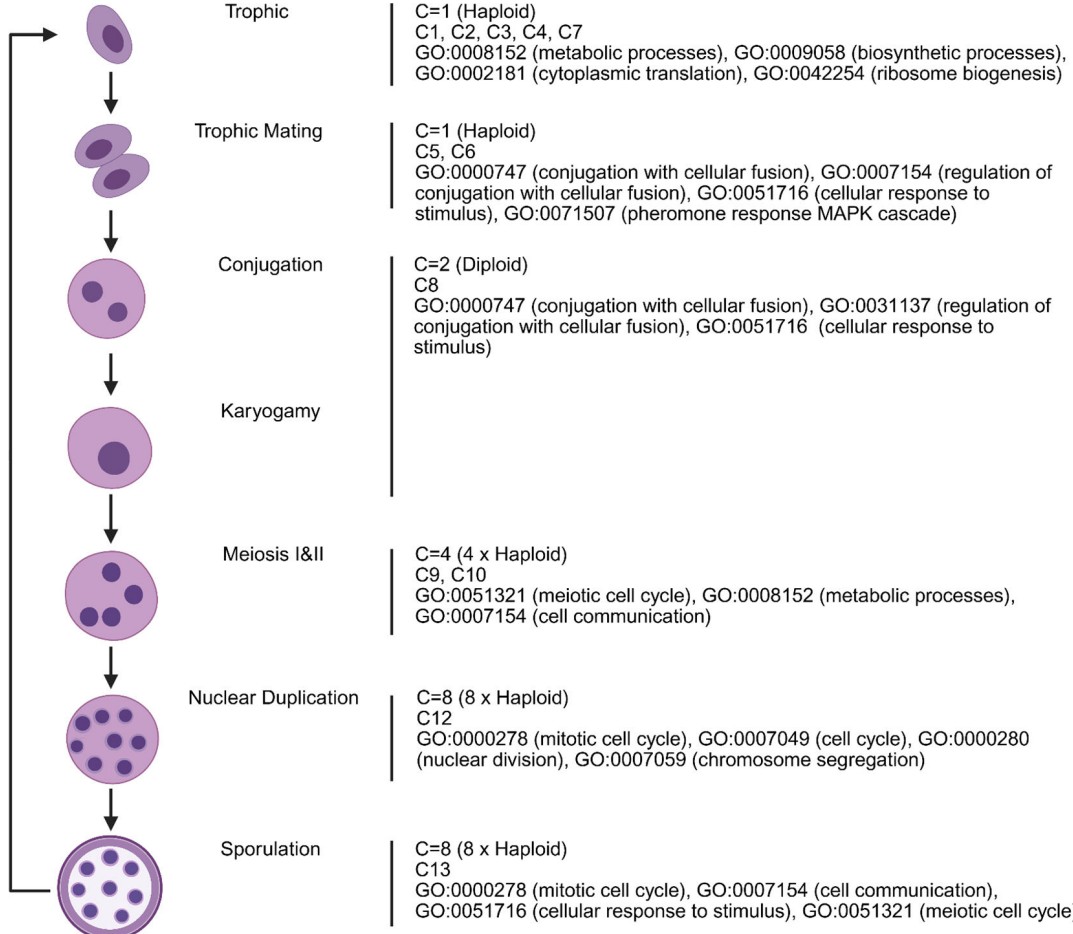

**FIG 4** The life cycle of *P. carinii* from scRNA seq trajectory analysis and GO enrichment. The figure illustrates the progression of the *P. carinii* life cycle, highlighting transcriptional clusters (C1–C13), associated ploidy levels (haploid, diploid, tetraploid, octoploid), and enriched gene ontology (GO) terms. Early clusters (C1–C5) represent biosynthetically active trophic forms (1C, haploid), characterized by upregulation of translation, ribosome biogenesis, and metabolic processes. Clusters C5 and C6 correspond to mating-competent haploid trophic cells, enriched for signaling and pheromone-related genes. Cluster C8, representing conjugation (2C, diploid), exhibits peak transcriptional complexity and marks sexual commitment. Karyogamy follows in C9, while C10 and C11 represent meiotic divisions (4C, tetraploid), enriched for meiotic and cell cycle processes. Cluster C12 corresponds to nuclear duplication (8C, octoploid), and C13 defines mature ascospore-containing asci, enriched for sporulation-related genes. This model suggests coordinated progression from trophic growth to sexual differentiation and ascus formation.

of nutrient-scavenging genes during mating, consistent with fungal sexual induction under starvation conditions, adds a novel layer to our understanding of metabolic reprogramming during sexual differentiation (21). This study, therefore, refines and expands traditional life cycle models of *Pneumocystis*, supporting an obligate sexual reproduction model driven by primary homothallism and a tightly regulated developmental progression.

Despite these advancements, limitations include the potential underrepresentation of adherent trophic forms lost during BALF collection (13). Future studies should optimize sampling methods to capture *P. carinii* adherent to host pneumocytes, as these forms likely exhibit distinct transcriptional profiles. A specific study using samples that retain host immune and epithelial cells would help clarify how cell–cell interactions influence fungal gene expression during the *P. carinii* life cycle. Multiple immune cell types, including neutrophils, macrophages, and lymphocytes, along with alveolar epithelial cells, play critical roles in shaping the *Pneumocystis* microenvironment and may drive stage-specific transcriptional changes (36–39). Expanding this approach to other species, such as *P. murina* or *P. jirovecii*, will further elucidate host adaptation strategies across

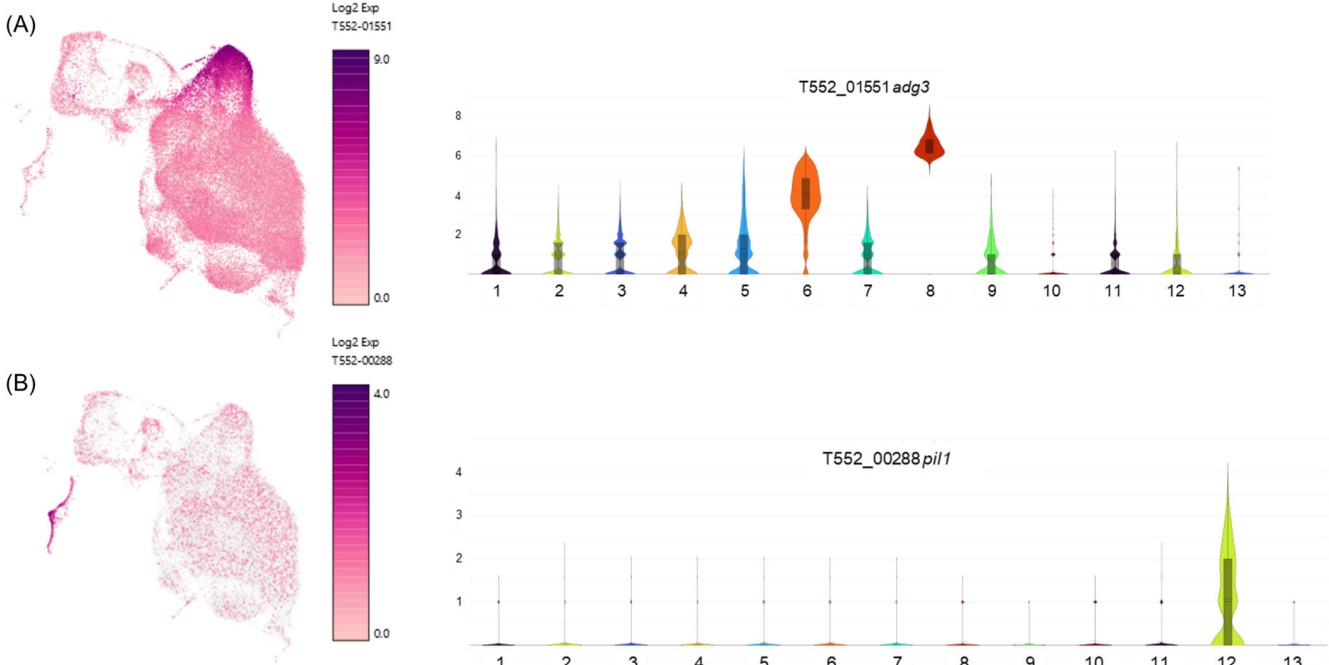

**FIG 5** Expression dynamics of *adg3* and *pil1* during transcriptional transitions from trophic growth to ascus maturation in *P. carinii*. (A) UMAP feature plot (left) and violin plot (right) of T552_01551 (*adg3*), a β-glucosidase gene predicted to encode a surface-localized hydrolase. *adg3* is moderately expressed in trophic clusters (C1–C4), sharply upregulated in mating-associated clusters (C5–C8), and repressed in reproductive stages (C10–C13). This expression pattern suggests roles in nutrient acquisition and cell wall remodeling during mating, with downregulation coinciding with ascus wall stabilization. (B) UMAP feature plot (left) and violin plot (right) of T552_00288 (*pil1*), a homolog of eisosome-associated membrane organizers involved in ascospore formation. *pil1* expression is specific to cluster C12, corresponding to stages of nuclear duplication and sporulation. These findings align with the role of *pil1* in membrane reorganization during ascospore encapsulation in other ascomycetes. Together, these gene expression profiles illustrate coordinated transcriptional transitions from trophic proliferation (C1–C4), through mating and fusion (C5–C8), to reproductive differentiation and ascus maturation (C9–C13).

*Pneumocystis* species. Overall, this study defines the transcriptional landscape of *P. carinii* across its life cycle, revealing distinct stage-specific gene expression programs and providing new insight into the regulation of sexual reproduction in an obligate fungal pathogen. These findings establish a framework for dissecting developmental transitions in host-associated fungi at single-cell resolution.

## MATERIALS AND METHODS

### Animal model for *P. carinii* pneumonia

To induce *P. carinii* pneumonia, male Sprague-Dawley rats (Charles River, Wilmington, MA), weighing 125–150 grams, were administered 20 mg/kg of Depo-Medrol (Pfizer, New York, NY) subcutaneously once per week for 8 weeks to induce immunosuppression. After 2 weeks of continuous immunosuppression, rats were inoculated intranasally with $1 \times 10^6$ *P. carinii* organisms suspended in PBS (9).

### Anidulafungin treatment for depletion of asci

Immunosuppression was maintained for 5 weeks post-inoculation. Rats were divided into two groups (*n* = 6 per group). One group received the anidulafungin (Eraxis, 5 mg/kg/day)(Pfizer, NY, NY) by intraperitoneal injection for an additional 3 weeks. The second group received no antifungal treatment. At the end of the 3 weeks of treatment period (8 weeks post-inoculation), all animals were euthanized, and the lungs were harvested. Microscopic enumeration of *P. carinii* cysts and trophic forms was performed

to confirm ascus depletion in the anidulafungin-treated group, as detailed in prior work by our lab (15).

## Enumeration of *P. carinii* trophic and ascus forms

*P. carinii* life cycle stages were quantified using established microscopy-based enumeration protocols. Three 10 μL aliquots were placed on microscope slides (Fisher Scientific, Pittsburgh, PA), air-dried, and heat-fixed. To distinguish developmental forms, asci were visualized using cresyl violet acetate (CVA; Sigma-Aldrich, St. Louis, MO), while total nuclei, including trophic and ascus forms, were stained using a Diff-Quik stain (Siemens Healthcare Diagnostics Inc., Newark, DE). Counts were averaged to determine the concentration and ratio of developmental stages (40). The final cell suspensions were normalized to a target concentration of 700 cells/μL for loading into the 10X Genomics single-cell RNA-seq assay.

## Isolation of *P. carinii* from rat lungs

*P. carinii* was extracted from rat lungs via bronchoalveolar lavage (BAL). To remove host cells, BAL fluids from three individual rats were pooled and sequentially filtered through 70 μm and 40 μm pore filters (Fisher Scientific, Pittsburgh, PA) (41). The filtrates were centrifuged at 2,500 × $g$ to pellet fungal cells, which were resuspended in 0.85% ammonium chloride solution at 37°C for 10 minutes to lyse red blood cells. After centrifugation, the resulting pellets were maintained in RPMI 1640 medium (Fisher Scientific, Pittsburgh, PA) supplemented with 20% fetal bovine serum (Cytiva, Sweden AB), MEM Non-Essential Amino Acids (Gibco, Pittsburgh, PA), MEM Vitamin Solution (Gibco, Pittsburgh, PA), penicillin-streptomycin (10,000 μg/mL,10,000 μg/mL; Gibco, Pittsburgh, PA), and vancomycin (5 mg/mL; Fisher Scientific, Pittsburgh, PA) (8). Following extractions, incubation at 37°C for 30 minutes in T25 tissue culture flasks facilitated host cell adherence and enabled the enzymatic degradation of extracellular DNA using DNase I (Thermo Fisher Scientific, Waltham, MA). DNase I was applied at a concentration of 100 U/mL to effectively remove DNA in cell culture systems (42). *P. carinii* organisms were collected from the supernatant, followed by centrifugation at 2,500 × $g$ to pellet and resuspend in 1 mL of 2% Ficoll-Hypaque before placement on a Ficoll gradient (17). The gradient was prepared with Ficoll (Ficoll: Millipore, Billerica, MD) concentrations diluted in 16% sodium diatrizoate (Hypaque: Sigma, St. Louis, MO), ranging from 4% to 12% increments. Following centrifugation at 300 × $g$, each layer was collected, and the cells were rinsed with cold Dulbecco's phosphate-buffered saline (DPBS; Fisher Scientific, Pittsburgh, PA) and pelleted by centrifugation at 2,500 × $g$. To minimize cell clumping, pellets were gently resuspended using a 20-gage ball-tip gavage needle (Becton Dickinson & Co., Franklin Lakes, NJ) before final centrifugation at 2,500 × $g$. The resulting samples were reconstituted in $Mg^{2+}$ and $Ca^{2}$-free RPMI 1640 medium (Fisher Scientific, Pittsburgh, PA) supplemented as described above.

## Single-cell library preparation and sequencing

Following the manufacturer's instructions, single-cell RNA sequencing (scRNA-seq) was performed using the Chromium Next GEM Single Cell 3' Reagent Kits v3.1 (Dual Index) from 10× Genomics. To facilitate the lysis of *P. carinii* asci, 1 μL of Zymolyase 100T (70 mg/mL; Asmbio, Cambridge, MA) was added to the gel bead mix before droplet generation. This enzymatic treatment enhances cell wall digestion, thereby improving the efficiency of in-droplet lysis during the scRNA-seq workflow (18). Briefly, cells were resuspended in the master mix and loaded into the Chromium chip with partitioning oil and gel beads to generate gel bead-in-emulsions (GEMs). Polyadenylated RNA underwent reverse transcription within each GEM, incorporating an Illumina TruSeq Read one primer sequence, a Unique Molecular Identifier (UMI), and a 10× Barcode. Post-GEM recovery, barcoded cDNA was purified using Silane DynaBeads and amplified via 14 cycles of PCR. The amplified cDNA was then enzymatically fragmented, size-selected,

adapter-ligated, and subjected to sample index PCR to construct sequencing libraries. Libraries were pooled and sequenced on an Illumina NovaSeq 6000 platform using an S4 flow cell with the following parameters: Read 1: 28 cycles; i7 Index: 10 cycles; i5 Index: 10 cycles; Read 2: 90 cycles.

## Single-cell RNA sequencing data processing and analysis

Raw sequencing reads from scRNA-seq experiments were processed using Cell Ranger v9.0 (10x Genomics). Reads were aligned to the *P. carinii* B80 reference genome with the removal of major surface glycoprotein genes (Accession: GCF_001477545.1) and only kept *P. carinii* B80 reads. Genes detected in fewer than 10 cells were excluded, and cell-level quality control (QC) filtering was applied using the following thresholds: UMI per cell (<400 and >2,300), genes per cell (<200 and >1,600), and gene expression complexity (log10[genes per UMI] ≤0.8) to remove low-quality or damaged cells that would result in technical noise. Expression data from all samples were merged and normalized using SCTransform and PCA using the top 50 principal components determined by an elbow plot (43). Clustering was conducted using the Louvain algorithm with a resolution parameter of 0.2, and visualization was performed using Uniform Manifold Approximation and Projection (UMAP) to represent transcriptional heterogeneity.

## Visualization and differential gene expression analysis using Loupe Browser

scRNA-seq data from *P. carinii* were analyzed using Loupe Browser v8.1.1 (10x Genomics) visualization (44). LoupeR converted Seurat v5.2.0 objects into a compatible format (10 x Genomics Software LoupeR, version 1.1.4). Data were projected using UMAP with clusters determined by Cell Ranger. Differential gene expression was assessed using the Significant Feature Comparison Analysis tool, selecting genes with an average occurrence of more than one count per barcoded spot, with expression values derived from $log_2$-transformed unique molecular identifiers (UMI) counts.

## Gene annotation and manual curation of *P. carinii* genes

*P. carinii* genes were annotated by mapping homologs to *S. pombe* genes (45). A custom BLAST database was generated from *S. pombe* protein sequences (Accession: GCF_000002945.1), against which *P. carinii* proteins (Accession: GCF_001477545.1) were aligned (46). Protein sequence comparisons were conducted using BLASTp on the public Galaxy server (version 22.05, usegalaxy.org) (47, 48). The threshold of the expected value (E-value) was set to <0.0001 to ensure high-confidence alignments. The top three matches with the highest bit scores were considered for annotation.

## GO term enrichment analysis and visualization

GO enrichment analysis was performed using g:Profiler (version e112_eg59_p19_25aa4782)(49, 50) and gene annotations were sourced from Ensembl (51) Fungi release. A significance threshold of adjusted $P < 0.05$ was applied, with multiple testing corrections performed using both the Bonferroni method and False Discovery Rate (FDR) adjustment.

GO term enrichment for biological and molecular processes was conducted using Cluster Profiler, with a background gene set derived from the *P. carinii* B80 genome (Accession: GCF_001477545.1) (52). Statistical significance was determined using Fisher's exact test with Benjamini-Hochberg correction ($P < 0.05$). Enrichment scores were $log_{10}$-transformed for visualization.

All visualization and statistical analysis were performed in Python 3.10 using a Google Colab environment (Notebook ID: GO_Pseudotime_ClusterViz_2024) (53, 54), heatmaps were generated with Seaborn v0.11.2, and line plots representing $-log_{10}$ adjusted $P$-values ($P < 0.05$) were produced using Matplotlib v3.5.3 and SciPy v1.10.1 (55, 56).

## Trajectory analysis

Trajectory analysis was conducted using Slingshot v2.14.0 to infer lineage structures within cell populations (57). Input data comprised UMAP components derived from Seurat-generated single-cell transcriptomic data. A minimum spanning tree (MST) was constructed to define global lineage relationships, and cell lineages were assigned using Slingshot's clustering-based approach. The root node was manually selected based on the most transcriptionally distinct cluster. Lineage-specific gene expression trends were analyzed using generalized additive models (GAMs) to identify genes that are dynamically regulated along inferred trajectories. Cell transitions and lineage progression were visualized using UMAP embedding, with cells ordered by the trajectory analysis.

## Marker gene validation by RT-qPCR

Candidate asci marker genes, T552_01968 (*mcp1*), T552_01932 (uncharacterized), and T552_01043 (*dmc1*), were selected for validation based on their high expression in Clusters 11–13 identified by scRNA-seq analysis. Three internal reference genes, thymidylate synthase T552_02292 (*TS*), T552_04188 (uncharacterized), and T551_00974 (uncharacterized), were selected based on stable expression and low cluster-to-cluster variance. The geometric mean of these three reference genes was used to normalize target gene expression (58, 59).

RNA was extracted using the Direct-zol RNA Miniprep Kit (Zymo Research). First-strand cDNA synthesis was performed using SuperScript IV VILO Master Mix (Thermo Fisher Scientific) and stored at –80°C until use. Primer efficiencies for all target and reference genes were determined using standard curves derived from serial dilutions, ranging from 98.3% to 100%, which validated the use of the $2^{-\Delta\Delta Ct}$ method. Gene-specific primers are listed in Table S4.

RT-qPCRs were run on an Applied Biosystems QuantStudio 3 Real-Time PCR System (Thermo Fisher Scientific, Waltham, MA) using PowerUp SYBR Green Master Mix (Thermo Fisher Scientific, Waltham, MA) under fast cycling conditions. Each 20 µL reaction contained 500 nM of each primer and was performed in three biological replicates with three technical replicates per sample. The fast thermocycling protocol consisted of 50°C for 2 minutes, followed by 95°C for 2 minutes, and then 40 cycles of 95°C for 1 second and 60°C for 20 seconds. Fluorescence was measured during the 60°C annealing/extension step, and melt curve analysis was performed to verify amplification specificity.

Normalized $\Delta Ct$ values were calculated using the geometric mean of the three reference gene Cts. Fold changes were determined using the $2^{-\Delta\Delta Ct}$ method, comparing untreated control samples to anidulafungin-treated populations depleted of asci.

## Statistical analysis

Unpaired two-tailed *t* tests were used to assess differences in gene expression between control and anidulafungin-treated groups in RT-qPCR validation experiments. For comparisons involving multiple target genes, *P* values were corrected for multiple testing using the Benjamini–Hochberg false discovery rate (FDR) method. For single-cell RNA-seq analysis, clustering was performed using graph-based Louvain modularity optimization. Differential gene expression was assessed using two approaches: (i) the Wilcoxon rank-sum test implemented in Seurat's functions after SCTransform normalization and (ii) the likelihood ratio test (LRT) based on a negative binomial generalized linear model, as implemented in the Loupe Browser (via 10× Genomics). GO term enrichment was assessed using hypergeometric overrepresentation analysis, with statistical significance determined by adjusted *P* values (FDR, Benjamini–Hochberg method). To prioritize biologically relevant terms, a composite score was calculated for each GO term as $\log_{10}$(fold enrichment) multiplied by $-\log_{10}$(adjusted *P* value), integrating both effect size and statistical strength. Composite scores were used for visualization and ranking in enrichment plots. All statistical analyses were performed using GraphPad Prism (v9.5.1) and R (v4.2.1). A *P* value of < 0.05 was considered statistically significant.

Descriptive statistics, including mean, standard deviation (SD), and standard error of the mean (SEM), were reported where appropriate.

## ACKNOWLEDGMENTS

The National Institutes of Health (NIH) R01HL146266 (M.T.C.) and the U.S. Department of Veterans Affairs (VA) 1I01B × 004441- 01 (M.T.C.) supported this work. M.T.C. is a senior research career scientist at the Cincinnati Veterans Affairs Medical Center.

## AUTHOR AFFILIATIONS

[1]Department of Internal Medicine, University of Cincinnati, College of Medicine, Cincinnati, Ohio, USA
[2]The Veterans Affairs Medical Center, Cincinnati, Ohio, USA
[3]Center for Autoimmune Genomics and Etiology, Cincinnati Children's Hospital Medical Center, Cincinnati, Ohio, USA
[4]Department of Pediatrics, University of Cincinnati, Cincinnati, Ohio, USA

## AUTHOR ORCIDs

Aaron W. Albee  http://orcid.org/0009-0008-5840-1890
Steven G. Sayson  http://orcid.org/0000-0002-7461-0704
Aleksey Porollo  http://orcid.org/0000-0002-3202-5099
Melanie T. Cushion  http://orcid.org/0000-0001-6621-2784

## FUNDING

| Funder | Grant(s) | Author(s) |
| --- | --- | --- |
| National Institutes of Health | R01HL146266 | Melanie T. Cushion |
| U.S. Department of Veterans Affairs | 1I01BX004441- 01 | Melanie T. Cushion |

## AUTHOR CONTRIBUTIONS

Aaron W. Albee, Conceptualization, Data curation, Formal analysis, Investigation, Methodology, Software, Validation, Visualization, Writing – original draft, Writing – review and editing | Steven G. Sayson, Conceptualization, Data curation, Formal analysis, Methodology, Project administration, Software, Supervision, Visualization, Writing – review and editing | Alan Ashbaugh, Formal analysis, Methodology, Writing – review and editing | Nicholas J. Wolf, Investigation, Methodology, Validation, Writing – review and editing | Aleksey Porollo, Data curation, Formal analysis, Methodology, Resources, Software, Visualization, Writing – review and editing | George Smulian, Conceptualization, Supervision, Writing – review and editing | Melanie T. Cushion, Conceptualization, Funding acquisition, Project administration, Resources, Supervision, Writing – review and editing

## DATA AVAILABILITY

scRNA-seq data have been deposited in the NCBI GEO database under accession number GSE300430. The deposited files include raw FASTQ files (paired-end) for each biological replicate; the cell-by-gene UMI count matrix (MTX format) and associated gene and barcode TSV files; metadata tables with cluster assignments and pseudotime annotations; and the SCTransform-normalized expression matrix used in downstream analyses.

## ADDITIONAL FILES

The following material is available online.

## Supplemental Material

**Fig. S1 (Spectrum01277-25-s0001.tif).** Optimization of separation, enrichment, and viability for single-cell RNA sequencing of *Pneumocystis carinii*.
**Fig. S2 (Spectrum01277-25-s0002.tif).** Viability and lysis data.
**Supplemental material (Spectrum01277-25-s0003.docx).** Supplemental methods; Tables S1 to S4.

## Open Peer Review

**PEER REVIEW HISTORY (review-history.pdf).** An accounting of the reviewer comments and feedback.

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
