## [Reviewer comments · Microbiology Spectrum]

Microbiology Spectrum

Single-Cell RNA Sequencing Defines Developmental Progression and Reproductive Transitions of *Pneumocystis carinii*

Aaron Albee, Steven Sayson, Nicholas Wolf, Alan Ashbaugh, Aleksey Porollo, A. George Smulian, and Melanie Cushion

Corresponding Author(s): Melanie Cushion, College of Medicine, Univ. of Cincinnati

Review Timeline:

Submission Date:	April 27, 2025
Editorial Decision:	May 30, 2025
Revision Received:	July 31, 2025
Accepted:	August 1, 2025

Editor: Kirsten Nielsen

Reviewer(s): Disclosure of reviewer identity is with reference to reviewer comments included in decision letter(s). The following individuals involved in review of your submission have agreed to reveal their identity: Alexandre Alanio (Reviewer #2)

Transaction Report:

DOI: <https://doi.org/10.1128/spectrum.01277-25>

Re: Spectrum01277-25 (**Single-Cell RNA Sequencing Defines Developmental Progression and Reproductive Transitions of *Pneumocystis carinii***)

Dear Prof. Melanie T. Cushion:

Thank you for the privilege of reviewing your work. Below you will find my comments, instructions from the Spectrum editorial office, and the reviewer comments.

Both Reviewers expressed excitement about the data and had only minimal concerns that were predominantly focused on the analysis methods. Please address the reviewer comments. I look forward to seeing the revised manuscript.

Revision Guidelines

Sincerely,
Kirsten Nielsen
Editor
Microbiology Spectrum

Reviewer #1 (Comments for the Author):

The paper by Albee conveys new information on scRNAseq in the *P. carinii* that also provides insight into replication and metabolism. The authors identified 13 clusters in the data and provided additional RNA velocity as gene ontology analysis. The paper provides novel insight into the life cycle of an important fungal pathogen. I only have minor comments.

1. The data were mapped to the *P. carinii* genome which is appropriate but it is possible that some of the reads/clusters are driven by intracellular organisms (despite the percoll). This could be analyzed in Cell Ranger by dual mapping to the rat genome and the fungal genome and determine if fungal reads are co-localizing with macrophage genes such as *Itgax*, *Cxc3cr1*, *Cd68*.
2. The first part of the results use *S. pombe* nomenclature and the qPCR studies use the T552 nomenclature. It would be helpful to add the *Sp* nomenclature to the qPCR sections in parentheses.
3. T552_02201 appears to be homologous with *Gas1* whereas as T552_02202 the *Bgs* family in *S. pombe*. Which clusters was T552_02202 up or down regulated in the Tables? Also, the tables are pictures, but it would be better if they were searchable such as a docx or xls file.

Reviewer #2 (Comments for the Author):

The authors studied single cell RNA sequencing upon isolation and enrichment of *P. carinii* obtained from rat BAL.

Methods : It is unclear how single cell experiments have been performed (automated sorter and dispenser ?).

As trophic form cells are sticky, how many cells have been sequenced ?

How the quality control regarding trophic form vs ascus proportion have been performed for the sequencing experiment ?

Why the author did rtqPCR and not single cell analysis after anidulafungin treatment ? It should have been interesting to see the differences with the same methodology.

Relative expression is not accurate using the $2^{-\Delta\Delta Ct}$ calculation. Each PCR target assay including the calibrator should be determined the efficiency. 3 calibrators should be selected and not only one. Read reference : Vandesompele J, Preter K de, Pattyn F, Poppe B, Roy N van, Paepe A de, Speleman F. Accurate normalization of real-time quantitative RT-PCR data by geometric averaging of multiple internal control genes. *Genome biology*. 2002 Jun 18;3(7):RESEARCH0034.

Response to Reviewers – ASM Spectrum Submission

Manuscript Title: Single-Cell RNA Sequencing Defines Developmental Progression and Reproductive Transitions of *Pneumocystis carinii*

Authors: Aaron Albee, Steven Sayson, Alan Ashbaugh, Nicholas J. Wolf, Aleksey Porollo, A. George Smulian, Melanie Cushion

Manuscript ID: Spectrum01277-25

Reviewer #1 Comments and Responses

GENERAL COMMENTS:

The paper by Albee conveys new information on scRNA-seq in *P. carinii* that also provides insight into replication and metabolism. The authors identified 13 clusters in the data and provided additional RNA velocity as gene ontology analysis. The paper provides novel insight into the life cycle of an important fungal pathogen. I only have minor comments.

Comment 1:

The data were mapped to the *P. carinii* genome, which is appropriate, but it is possible that some of the reads/clusters are driven by intracellular organisms (despite the Percoll). This could be analyzed in Cell Ranger by dual mapping to the rat genome and the fungal genome and determine if fungal reads are co-localizing with macrophage genes such as *Itgax*, *Cx3cr1*, *Cd68*.

Response:

This is an insightful point. While dual mapping to both *Rattus norvegicus* and *P. carinii* is technically feasible, our upstream enrichment protocol was designed to deplete host mammalian cells. We used adherence-based separation (Lines 417–429) and Ficoll gradient centrifugation (Supplementary Figure S1B) to remove large host cells. As a result, host cells were underrepresented, and dual mapping would not provide a representative analysis of the co-encapsulation of host and pathogen. A future study designed to retain host immune populations would be better suited for such analysis. (Manuscript Lines 370-376).

Comment 2:

The first part of the results uses *S. pombe* nomenclature, and the qPCR studies use the T552 nomenclature. It would be helpful to add the *S. pombe* nomenclature to the qPCR sections in parentheses.

Response:

This suggestion has been implemented. We now include *S. pombe* ortholog names in parentheses alongside T552 identifiers throughout the qPCR Results and in the Figure 3 legend to aid cross-referencing (Manuscript Lines 286-292 and 580-589)

Comment 3:

T552_02201 appears to be homologous with *Gas1*, whereas T552_02202 is part of the *Bgs* family in *S. pombe*. Which clusters were T552_02202 up- or downregulated in the

Tables? Also, the tables are pictures, but it would be better if they were searchable, such as a docx or xls file.

Response:

T552_02202 (*gsc1*), a homolog of *bgs1* in *S. pombe* encodes the catalytic subunit of the β -1,3-glucan synthase complex. It is significantly upregulated in Clusters 11–13 based on both fold change and adjusted *p*-value, though it does not fall within the top 25 differentially expressed genes. This is now described in the Discussion (Manuscript Lines 326-332). Additional β -glucan biosynthesis genes, including those from the *gas*, *bgl*, and *bgs* families, are upregulated in Clusters 12 and 13. These patterns support a temporally regulated transition from substrate acquisition to ascus wall synthesis. All image-based tables have been replaced with formatted text tables (Manuscript line 624-629).

Reviewer #2 Comments and Responses

GENERAL COMMENTS:

The authors studied single-cell RNA sequencing upon isolation and enrichment of *P. carinii* obtained from rat BAL.

Comment 1:

It is unclear how single-cell experiments have been performed (e.g., automated sorter and dispenser).

Response:

To clarify, single-cell partitioning was performed using the 10X Genomics Chromium Controller, which encapsulates individual cells into gel bead-in-emulsion (GEM) droplets via a microfluidic platform. We have moved this description from the Supplementary Methods into the Manuscript Methods section for transparency (Manuscript Lines 432–457).

Comment 2:

As trophic-form cells are sticky, how many cells have been sequenced?

Response:

A total of 87,502 individual *P. carinii* cells were recovered and sequenced. This value is now included in the Results (Manuscript Lines 104–106), with sample-specific numbers detailed in Supplementary Table ST3.

Comment 3:

How has quality control regarding trophic form vs. ascus proportion been performed for the sequencing experiment?

Response:

Stage composition was quantified via microscopic counts of BALF samples prior to single-cell loading. These assessments confirmed the enrichment of asci, as well as sample-specific ratios of both trophic and ascus cells, and are now shown in Supplementary Figure S1 and described in the Results (Manuscript Lines 85–97).

Comment 4:

Why did the authors do RT-qPCR and not single-cell analysis after anidulafungin treatment?

Response:

This is an excellent question. Our primary objectives in this study were to reconstruct the untreated developmental trajectory of *P. carinii* and define transcriptional signatures (marker genes) associated with distinct life cycle stages. To establish a baseline reference transcriptome, we selected a single time point at the end stage of infection. RT-qPCR was then used to validate stage-specific marker genes in parallel samples, where asci were selectively depleted using established echinocandin protocols. This allowed us to test whether the expression of late-stage markers decreased in the absence of mature asci, confirming their developmental specificity. scRNA seq analysis of treated sample groups is a valuable direction for future work, and we are actively pursuing this in follow-up studies.

Comment 5:

Relative expression is not accurate using the $2^{-\Delta\Delta Ct}$ calculation. Three calibrators should be selected, not just one. Read reference: Vandesompele J, Preter K de, Pattyn F, Poppe B, Roy N van, Paepe A de, Speleman F. Accurate normalization of real-time quantitative RT-PCR data by geometric averaging of multiple internal control genes. *Genome Biology*. 2002 Jun 18;3(7):RESEARCH0034.

Response:

We appreciate the reviewer's suggestion regarding our selection criteria for RT-qPCR reference genes. To clarify, we assessed both raw and normalized single-cell RNA-seq data to identify suitable internal controls. Specifically, we selected T552_04188 (uncharacterized) and T552_00947 (uncharacterized) based on consistently high raw expression across most cells in the dataset. Raw UMI counts for both genes were well above the dataset median, indicating strong transcriptional activity and reliable detection.

Following normalization using SCTransform, we confirmed that both genes maintained stable expression across all transcriptional clusters (C1–C13). Neither gene showed significant dropout or restriction to a specific developmental stage. Uniformity was evaluated by visualizing violin plots, and statistical testing confirmed that neither gene exhibited significant differential expression between clusters (adjusted $P \approx 1$). Together, these criteria support the selection of T552_04188 and T552_00947 as suitable reference genes for geometric averaging, in accordance with the method described by Vandesompele et al. (2002).

To ensure accurate relative quantification, we empirically determined primer efficiencies for all target and reference genes following the best practices outlined by Vandesompele et al. (2002). Standard curves were generated from 5-point, 10-fold serial dilutions of cDNA, and efficiencies ranged from 98.3% to 100%. These values support the use of the $2^{-\Delta\Delta Ct}$ method under conditions of near-optimal amplification.

Expression values were normalized to the geometric mean of the three reference gene Ct values to minimize normalization bias and improve comparability across samples, as

recommended by Vandesompele et al. (2002). This approach was further supported by the extended ΔC_t method for treatment–control comparisons described by Riedel et al. (2014). These methodological updates are now reflected in the results section (Manuscript lines 282-301), the revised Methods section (Manuscript lines 502–525), and Figure 3 (Manuscript lines 581-596),

Re: Spectrum01277-25R1 (**Single-Cell RNA Sequencing Defines Developmental Progression and Reproductive Transitions of *Pneumocystis carinii***)

Dear Prof. Melanie T. Cushion:

Authors appropriately address all the reviewer concerns

Your manuscript has been accepted, and I am forwarding it to the ASM production staff for publication. Your paper will first be checked to make sure all elements meet the technical requirements. ASM staff will contact you if anything needs to be revised before copyediting and production can begin. Otherwise, you will be notified when your proofs are ready to be viewed.

Sincerely,
Kirsten Nielsen
Editor
Microbiology Spectrum